# The Surprising Effectiveness of SP Voting with Partial Preferences

**Hadi Hosseini**
College of Information Sciences and Technology
Penn State University, USA
hadi@psu.edu

**Debmalya Mandal**
Department of Computer Science
University of Warwick, UK
Debmalya.Mandal@warwick.ac.uk

**Amrit Puhan**
College of Information Sciences and Technology
Penn State University, USA
avp6267@psu.edu

## Abstract

We consider the problem of recovering the ground truth ordering (ranking, top-$k$, or others) over a large number of alternatives. The wisdom of crowd is a heuristic approach based on Condorcet's Jury theorem to address this problem through collective opinions. This approach fails to recover the ground truth when the majority of the crowd is misinformed. The *surprisingly popular* (SP) algorithm [36] is an alternative approach that is able to recover the ground truth even when experts are in minority. The SP algorithm requires the voters to predict other voters' report in the form of a full probability distribution over all rankings of alternatives. However, when the number of alternatives, $m$, is large, eliciting the prediction report or even the vote over $m$ alternatives might be too costly. In this paper, we design a scalable alternative of the SP algorithm which only requires eliciting partial preferences from the voters, and propose new variants of the SP algorithm. In particular, we propose two versions—*Aggregated-SP* and *Partial-SP*—that ask voters to report vote and prediction on a subset of size $k$ ($\ll m$) in terms of top alternative, partial rank, or an approval set. Through a large-scale crowdsourcing experiment on MTurk, we show that both of our approaches outperform conventional preference aggregation algorithms for the recovery of ground truth rankings, when measured in terms of Kendall-Tau distance and Spearman's $\rho$. We further analyze the collected data and demonstrate that voters' behavior in the experiment, including the minority of the experts, and the SP phenomenon, can be correctly simulated by a concentric mixtures of Mallows model. Finally, we provide theoretical bounds on the sample complexity of SP algorithms with partial rankings to demonstrate the theoretical guarantees of the proposed methods.

## 1 Introduction

The wisdom of the crowds is a systematic approach to statistically combine the opinions of a diverse group of (non-expert) individuals to achieve a final collective truth. It dates back to Sir Francis Galton's observation—based on Aristotle's hypothesis—that the point estimation of a continuous value using the noisy individual opinions can recover its true value with high accuracy [22]. In the modern era, the wisdom of the crowd has been the foundation of legal, political, and social systems with the premise that one can recover the truth by collecting the opinion of a large number of diverse individuals (e.g. trial by jury, election polling, and Q&A platforms such as Reddit/Quora).

38th Conference on Neural Information Processing Systems (NeurIPS 2024).

The formal arguments of this phenomenon—rooted in social choice theory—is provided by *Condorcet's Jury Theorem* [20], which states that under the condition of independent opinions and each individual having more than a $50\%$ chance of selecting the correct answer, the probability of the majority decision being correct increases as the size of the crowd increases. However, this approach may fail when the majority of the crowd are misinformed (less than 50% chance of selecting the correct answer) [19] or are systematically biased [46]. In other words, when the experts are in minority, simply aggregating individuals' opinions (regardless of the aggregation method) cannot recover the truth.

To overcome this challenge, Prelec et al. [36] proposed a simple, yet effective, method called the *Surprisingly Popular* (SP) algorithm, which is able to uncover the ground truth even when the majority opinion is wrong. The approach works by asking each individual about their opinion (the *vote*) along with an additional *meta-question* to predict the majority opinion of other individuals (the *prediction*). The surprisingly popular algorithm then selects an answer whose actual frequency in the votes is greater than its average predicted frequency, and it will provably recover the correct answer with probability 1, as the number of individuals grows in the limit, even when experts are in minority.

While the SP algorithm is effective in estimating a continuous value (e.g. the value of a painting) or a binary vote (e.g. "Is São Paulo the capital of Brazil?"), it cannot be directly applied to recover true ordinal rankings over a set of $m$ alternatives due to the large number of votes ($m!$), and more importantly, eliciting predictions over a complete rankings. Hosseini et al. [25] extended this approach to rankings by proposing an algorithm, called *Surprisingly Popular Voting*, that can accurately recover the ground-truth ranking over multiple alternatives by eliciting a complete ranking as a vote and only a single majority prediction (as opposed to full probability distributions over $m!$ rankings).[1] Despite its success in finding the ground-truth ranking over a small number of alternatives, it remains unclear how to adapt it to settings with large number of alternatives where only partial preferences (e.g. pairwise comparisons or partial ranks) can be elicited. Thus, it raises the following questions:

> *How can we design scalable algorithms based on the surprisingly popular method that recovers the ground truth only by eliciting partial preferences from voters? What elicitation formats and aggregation algorithms are more effective in recovering the full ranking over all alternatives?*

**Our Contributions.** We focus on developing methods, based on the surprisingly popular approach, that *only* elicit partial vote and prediction information to find the full ranking. Given a set of $m$ alternatives, we ask individuals to provide their rank-ordered vote and predictions on a subset of size $k \, (\ll m)$ of alternatives. Informally, we ask them to identify the most preferred alternative among the $k$ choices (Top), select the $t < k$ most preferred alternatives with no order (Approval($t$)), or provide a rank-ordered list of all $k$ alternatives (Rank). The precise formulation is provided in Section 2.1.

Given that the SP algorithm [36] and its extension to rankings [25] do not generalize to partial preferences with large number of alternatives, we design two novel aggregation methods, namely Partial-SP and Aggregated-SP algorithms. On a high level, these algorithms use a carefully crafted method to select subsets of size $k \ll m$ for vote and prediction elicitation, and apply the SP method either independently on each subset (Partial-SP) or on the aggregated (potentially partial) votes and ranks (Aggregated-SP).

We conduct a human-subject study with 432 participants recruited from Amazon's Mechanical Turk (MTurk) to empirically evaluate the performance of our SP algorithms using metrics such as the *Kendall-Tau distance* from the full ground truth ranking and *Spearman's rank correlation* coefficient. We consider several classical vote aggregation methods (e.g. Borda, Copeland, Maximin, Schulze) as benchmarks—rooted in the computational social choice theory—that operate solely on votes (and not prediction information). Our results show that the SP voting algorithms perform significantly better than the classical methods when the vote and prediction information only contain partial rankings. We also observe that SP voting algorithms are effective even when restricted to approval votes.

Moreover, we demonstrate that voters' behavior in the experiment, including the minority of the experts can be correctly simulated by a concentric mixtures of Mallows model [33, 10]. Finally, we provide theoretical bounds on the sample complexity of the SP algorithms with partial preferences to

---

[1]This paper also explored various elicitation techniques combining vote and prediction questions based on only top choice and complete rankings.

further demonstrate the theoretical guarantees of the proposed methods. We show that the sample complexity only depends on the size of the subset $k$ which is significantly smaller than $m$.

## 1.1 Related Work

Our work is related to *information elicitation*, and *partial aggregation* which we discuss briefly.

**Information Elicitation**. Various information elicitation schemes [35, 36, 48, 17] attempt to incentivize voters to reveal useful information, often through the investment of efforts. Our work is primarily related to the *surprisingly popular algorithm* [36] which is a a novel second-order information based elicitation scheme. This framework has since been used to incentivize truthful behaviour in agents [35, 42, 43], mitigate biases in academic peer review [32], elicit expert knowledge [27], and aggregate information [9]. Our study builds upon this literature, specifically addressing the challenges in rank recovery. Originally, the SP algorithm by Prelec et al. [36] required data on all $m!$ potential rankings for $m$ alternatives , a requirement that becomes impractical as $m$ increases. Hosseini et al. [25] addressed this by developing a Surprisingly Popular Voting algorithm that leverages pairwise preference data across $\binom{m}{2}$ alternatives. This approach doesn't scale when $m$ is large, and our contribution lies in advancing this methodology by proposing a scalable generalization of the Surprisingly Popular Voting method with partial preferences.

**Partial Aggregation**. In situations where it is difficult or not necessary to elicit complete rankings from voters, partial preferences are used. Partial vote aggregation has different solution concepts [6]. Partial preferences can be used to conclude which alternatives are necessary and possible winners based on the preference profiles [26, 14, 47, 2, 3, 49]. The primary goal of partial aggregation methods is to either minimize the amount of information communicated by the voters [12, 45, 40] or to reduce the number of queries that each voter needs to answer [37, 14]. Our work attempts to reduce such communication from the voters by eliciting partial preferences.

More broadly, our work is also related to *information aggregation*, and *probabilistic rank-order models* which we discuss further in Appendix A.

## 2 Model

In this section, we formally define the model for Surprisingly Popular Voting in the context of partial preferences. Let $A = \{a_1, a_2, ..., a_m\}$ denote the set of $m$ possible alternatives. The set $\mathcal{L}(A)$ represents all possible complete rankings over the alternatives. Let $\sigma \in \mathcal{L}(A)$ represent a complete ranking of the $m$ possible alternatives. We denote the ground truth ranking by $\pi^\star \in \mathcal{L}(A)$; which is assumed to be drawn from a prior $P(\cdot)$ over $\mathcal{L}(A)$. Voter $i$ observes a ranking $\pi_i$ that is assumed to be a noisy version of the ground truth ranking $\pi^\star$. We will write $\mathrm{Pr}_s(\pi_i \mid \pi^\star)$ to denote the probability that the voter $i$ observes her ranking $\pi_i$ given the ground truth $\pi^\star$.

Given voter $i$'s ranking $\pi_i$ and the prior $P(\cdot)$, voter $i$ can compute the posterior distribution over the ground truth using the Bayes rule.

$$\mathrm{Pr}_g(\pi^\star \mid \pi_i) = \frac{\mathrm{Pr}_s(\pi_i|\pi^\star)\cdot P(\pi^\star)}{\sum_{\pi' \in \mathcal{L}(A)} \mathrm{Pr}_s(\pi_i|\pi')\cdot P(\pi')} \tag{1}$$

Using the posterior over the ground truth, voter $i$ can also compute a distribution over the rankings observed by another voter.

$$\mathrm{Pr}_o(\pi_j \mid \pi_i) = \sum_{\pi' \in \mathcal{L}(A)} \mathrm{Pr}_s(\pi_j \mid \pi') \cdot \mathrm{Pr}_g(\pi' \mid \pi_i) \tag{2}$$

The *surprisingly popular algorithm* [36] asks voters to report their votes, and posterior over others' votes. For each ranking $\pi'$, it then computes the frequency $f(\pi') = \frac{1}{n}\sum_i \mathbf{1}[\pi = \pi']$, and posterior $g(\pi \mid \pi') = \frac{1}{|\{i:\pi_i=\pi'\}|}\sum_{i:\pi_i=\pi'} \mathrm{Pr}_o(\pi \mid \pi_i)$, and finally picks the ranking with highest *prediction normalized votes*.[2]

$$\widehat{\pi} \in \mathrm{argmax}_\pi \overline{V}(\pi) = f(\pi) \cdot \sum_{\pi' \in \Pi} \frac{g(\pi'|\pi)}{g(\pi|\pi')} \tag{3}$$

As observed by Hosseini et al. [25], eliciting full posterior and even the vote might be prohibitive if the number of alternatives $m$ is huge. In this work, we are concerned about eliciting partial rankings

---

[2]This is the direct application of SP algorithm [36] by considering $m!$ possible ground truths.

over subsets of size $k \ll m$. Let us fix a subset $T \subseteq A$ of size $k$. Then the probability of a partial ranking $\sigma_i$ is given as

$$\Pr_s(\sigma_i \mid \pi^\star) = \sum_{\pi : \pi \triangleright \sigma_i} \Pr_s(\pi \mid \pi^\star) \tag{4}$$

Here we use the notation $\pi \triangleright \sigma_i$ to indicate that the ranking $\pi$ when restricted to the set $T$ is $\sigma_i$.

We can also naturally extend definition 1 to define the posterior distribution over partial preferences given a partial preference $\sigma_i$. In order to do so, let us first define the posterior over full ground truth $\pi^\star$ given $\sigma_i$ as,

$$\Pr_g(\pi^\star \mid \sigma_i) = \frac{\Pr_s(\sigma_i \mid \pi^\star) P(\pi^\star)}{\sum_{\tilde{\pi}} \Pr_s(\sigma_i \mid \tilde{\pi}) P(\tilde{\pi})}$$

where one can use definition (4) to compute $\Pr_s(\sigma \mid \pi^\star)$. Now we can write down the posterior probability over the partial ground truth $\tilde{\sigma}$ as follows.

$$\Pr_g(\tilde{\sigma} \mid \sigma_i) = \sum_{\pi : \pi \triangleright \tilde{\sigma}_i} \Pr_g(\pi \mid \sigma_i) \tag{5}$$

Finally, we can write the posterior over another partial ranking $\sigma'$ over the subset $T$.

$$\Pr_o(\sigma' \mid \sigma_i) = \sum_{\tilde{\sigma}} \Pr_g(\tilde{\sigma} \mid \sigma_i) \Pr_s(\sigma' \mid \tilde{\sigma}) \tag{6}$$

In this work, our aim is to propose several versions of surprising popular algorithm that work with partial preferences. As shown in definition 3, it requires eliciting information regarding voters partial preferences, and posterior over others' partial preferences (as defined in eq. (6)). Next, we discuss various ways of eliciting such information from the voters.

## 2.1 Elicitation Formats

Given a subset of size $k \ll m$ alternatives, voter $i$'s prediction $\Pr_o(\cdot \mid \sigma_i)$ is a distribution over $k!$ rankings. In practice, this renders elicitation of full prediction information difficult, if not impossible, due to its cognitive overload. Thus, we focus on simple, and more explainable, elicitation methods that rely on ordinal information either by identifying the most preferred alternative (`Top`), selecting the most preferred $t$ alternatives (`Approval(t)`), or a complete ranking of the partial set (`Rank`). The formal definitions can be found in Appendix B.

Given these elicitation methods, we study different combinations of formats for votes and predictions where the first component indicates the vote format and the second component denotes the prediction format. These give rise to nine formats: `Top-None`, `Top-Top`, `Top-Approval(t)`, `Top-Rank`, `Approval(t)-Rank`, `Approval(t)-Approval(t)`, `Rank-None`, `Rank-Top`, and `Rank-Rank`. For `Approval(t)`, the approval set of size $t \in \{1, 2, 3\}$ is selected. Note that `Approval(1)` $\equiv$ `Top`.

## 3 Aggregation Algorithms for Partial Preferences

The surprisingly popular method (as we discussed in section 1) cannot be applied directly to find a full ranking using only partial preferences. Thus, we develop two vote aggregation algorithms that *only* rely on partial ordinal preferences both for votes and predictions. On the high level, the two algorithms differ on how and when they implement the SP method, whether independently on each subset (Partial-SP) or on the aggregated (potentially partial) votes and ranks (Aggregated-SP). Here we provide a high-level description for each of the algorithms; additional details and exact pseudo-codes are relegated to Appendix D.

**Partial-SP.** The key element of this algorithm is utilizing SP voting on the partial rankings obtained at each step. It takes a set of potentially overlapping subsets of alternatives and a voting rule as input and proceeds as follows: For each subset $S_j$ of alternatives, collect votes and predictions from voters on this subset according to one of the elicitation formats detailed in Section 2.1. Compute the ground truth partial ranking on the subset $S_j$ using the SP algorithm. Aggregate all partial rankings using a voting rule (e.g. Condorcet) to find a full ranking over all alternatives (breaking ties at random).

**Aggregated-SP.** This variation utilizes SP voting on the final rankings over votes and predictions. It takes a set of potentially overlapping subsets of alternatives and a voting rule as input and proceeds as follows: For each subset $S_j$ of alternatives, collect votes and predictions from voters on this subset according to one of the elicitation formats detailed in Section 2.1. Aggregate all votes (partial

rankings) using a voting rule (e.g., Condorcet) to find the aggregated vote over all alternatives, breaking ties at random. Predictions are not aggregated to preserve conditional prediction information crucial for SP voting. Apply SP algorithm pairwise across all alternatives where for each pair $(a, b)$, the vote information is derived from the scores of $a$ and $b$ based on the aggregation rule used, and the prediction information is used to find the conditional probabilities, $P(a|b)$ and $P(b|a)$. Breaking ties at random throughout this process results in a full ranking over all alternatives.

**Subset Selection.** The algorithms described in this section rely on partial rankings on the subset of alternatives. Given $m$ alternatives, we carefully select subsets of size $k$ with an inter-alternative pairwise distance of $s$ between elements from the ground-truth ranking. Formally, a subset $S_j$ of size $k$ is generated as follows:

$$S_j = \{a_{1+j}, a_{1+j+s}, \ldots, a_{1+j+(k-1)s}\}, \tag{7}$$

where $j \geq 0$ and $j + (k-1)s < m$, ensuring elements are within the range of $m$ alternatives. We get a total of $m - (k-1)s$ subsets. Note that we use overlapping subsets so as to introduce transitivity among different subsets enabling us to compare alternatives across different subsets. This leads to an improvement in the accuracy of our algorithms as we discuss in Section 5.

# 4 Experimental Design

This section describes the experimental design of the Amazon Mechanical Turk (MTurk) study to assess the comparative efficacy of Partial-SP and Aggregated-SP against other voting rules for partial preferences. Participants in this study were asked to answer a series of questions, wherein they were required to express their preferences by voting on a range of alternatives. In addition to casting their own votes, participants were asked to predict the collective preference of others for the same set of alternatives. Data was collected from 432 respondents. Each participant was given a 20-minute window to complete a series of 18 questions (see details below).[3] **Datasets.** The survey encompassed three distinct domains: (i) The *geography* dataset contains 36 countries with their population estimates, according to the United Nations, (ii) The *movies* dataset contains 36 movies with their lifetime box-office gross earnings, and (iii) The *paintings* dataset contains 36 paintings with their latest auction prices.[4]

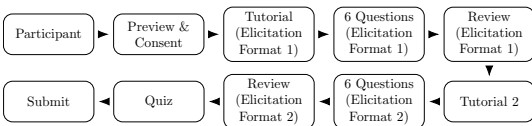

Figure 1: Workflow of a participant

**Questions.** We explored 36 alternatives per domain, aiming to gather partial preferences from voters. Each question featured a subset of alternatives, with the size of each subset maintained uniformly throughout the experiment.

Each participant was presented with a subset of 5 alternatives, selected based on an inter-alternative gap of 6 positions within the ground-truth ranking. This strategy was designed to balance the cognitive load against the quality of the responses. We tested subset sizes of 4 to 6 and inter-alternative gaps of 3 to 8, finding that larger sizes and wider gaps generally enhanced ground-truth recovery. However, larger subset sizes increase cognitive load for participants, and wider gaps reduce overlap between subsets when limited to 36 alternatives. For each combination of 12 subsets, 9 elicitation formats, and 3 domains, each question received 16 responses.

The survey was structured for each participant to answer two questions from each of the three domains and two elicitation formats, totaling 12 questions per participant. Figure 1 shows the workflow for each participant; each participant was assigned 18 questions to answer. Refer to Appendix E for details on the tutorials, participant qualifications for the MTurk study, and review questions regarding the perceived difficulty and expressiveness of the study.

**Elicitation Formats.** We use various elicitation formats (as described in section 2.1). For example, consider a question that requires participants to rank five movies: a) Rogue One: A Star Wars Story, b) Titanic, c) Toy Story 3, d) The Dark Knight Rises, and e) Jumanji: Welcome to the Jungle—based on their lifetime gross earnings. Under the `Approval(3)-Rank` elicitation format, the structure of the vote and prediction questions would be framed as follows:

---

[3]This study received IRB approval from the ethics board, which is available upon request.

[4]The dataset can be found here -https://github.com/amrit19/Surprisingly-Popular-Voting-Partial

- **Part A (vote):** *"Which among the following movies are the top three in terms of highest grossing income of all time?"*
- **Part B (prediction):** *"Considering that other participants will also respond to Part A, in what order do you predict the following movies will be ranked, from the most common response (top) to the least common (bottom)?"*

Refer to Appendix I for further details about formulations of all nine elicitation formats, the consent form, the tutorial for each domain, screenshots, and other details.

## 5   Results and Analysis

In this section, we present the results of this study averaged across all three domains. We measure the accuracy of the proposed SP algorithms (Partial-SP and Aggregated-SP) in predicting the full ground-truth ranking, in comparison with common vote aggregation methods (e.g. Borda, Copeland, Maximin, Schulze). The details of these aggregation methods is provided in Appendix C.

Additionally, we compare the elicitation formats (described in Section 2.1) with respect to cognitive effort (measured by response time and difficulty) and expressiveness (measured directly by survey questions). They are provided in Appendix G.2

### 5.1   Accuracy Metrics

To capture the error in predicting the full ground-truth ranking, we use three different metrics: (i) the *Kendall-Tau* correlation, which measures the distance between ordinal rankings, (ii) *Spearman's* $\rho$ correlation, which measures the statistical dependence between ordinal rankings, (iii) *Pairwise hit rate*, which measures the fraction of pairs at distance $d$ that are correctly ranked with respect to the ground-truth ranking, and (iv) *Top-$t$ hit rate*, which measures the fraction of alternatives that are predicted correctly (in no order) in most preferred $t$ compared to the ground-truth ranking. The formal definitions can be found in Appendix G.1.

For example, consider the ground-truth ranking $a \succ b \succ c \succ d$. The predicted ranking $b \succ a \succ d \succ c$ has a pairwise hit rate of $1/3$ at distance 1, 1 at distances 2 and 3. Its Top-1 hit rate is 0, Top-2 is 1, Top-3 is $2/3$, and Top-4 is 1.

### 5.2   Predicting the Ground Truth Ranking

Figure 2 illustrates the performance of SP algorithms measured by Kendall-Tau and Spearman's correlations. We fix Copeland as the aggregation rule used in both variations of SP voting and compare the accuracy with applying Copeland on votes alone (without the use of prediction information).

**Statistical correlations and elicitation.** SP voting produces rankings with a significantly higher correlation with the ground truth ranking, and this effect improves as the information provided as votes and prediction becomes more expressive. In particular, `Rank-Rank` and `Approval(3)-Rank` outperform all other elicitation formats. We note that Aggregated-SP seem to be more reliant on the vote information, compared to the predictions, as it can seen in `Top-Rank` vs. `Rank-Top`. In contrast, Partial-SP does not exhibit any significant favor for vote vs. prediction information as both `Top-Rank` and `Rank-Top` improve by additional information. However, the difference between them is not statistically significant.

Interestingly, eliciting unordered information for both noisy votes and predictions (e.g. `Approval(2)-Approval(2)`) seem to be sufficient in recovering the ground truth—raising the question of whether pairwise comparisons are necessary in designing SP algorithms.

**Hit rates.** With respect to pairwise hit rate and the Top-$t$ hit rate, the noisy prediction information significantly improves the performance of the Partial-SP algorithm (with lower variance) as shown in Figure 3. The results for Aggregated-SP are qualitatively similar and are presented in Appendix G.5. The slight dip in pairwise hit rate, can be explained by the survey's design choice of an inter-alternative distance of 6, leading to fewer comparisons being available for these pairs.

**Partial-SP vs. Aggregated-SP.** While both variants of the SP algorithm significantly outperform common voting rules by utilizing (noisy) prediction information, the Partial-SP algorithm significantly outperforms the Aggregated-SP algorithm (see Figure 2 and Figure 4). This could be explained

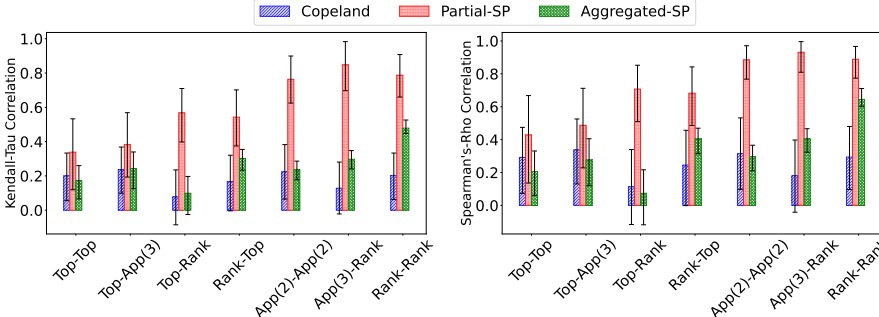

Figure 2: Comparing the predicted and ground-truth rankings for different elicitation formats using Kendall-Tau and Spearman's $\rho$ correlations (higher is better). All results use Copeland as their aggregation rule.

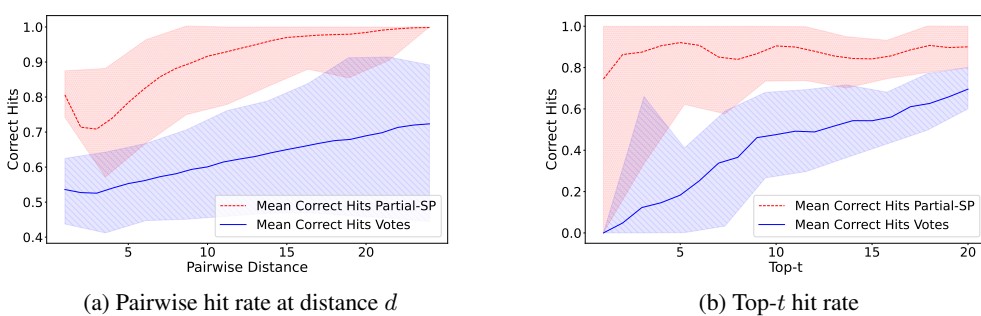

(a) Pairwise hit rate at distance $d$        (b) Top-$t$ hit rate

Figure 3: Comparing the Partial-SP algorithm with Copeland (no prediction information) measured by pairwise and Top-$t$ hit rates. The elicitation format is `Approval(2)-Approval(2)`.

by the importance of 'correcting' noisy votes on the subsets of alternatives because the prediction information of the Partial-SP algorithm helps identify experts early on in predicting partial rankings of these alternatives.

For Partial-SP, the plots indicate no statistical significance between `Approval(2)-Approval(2)`, `Approval(3)-Rank`, and `Rank-Rank` elicitation formats, suggesting they perform as well as `Rank-Rank`. An interesting ramification here is demonstrating that approval sets not only perform well in predicting the ground truth, but also pose less cognitive burden on voters compared to those elicitation formats that ask for rankings (see appendix G.2).

**Domain impacts.** Performance of Partial-SP and Aggregated-SP is robust across domains. They outperform common voting rules with the sole exception of the Schulze method, which matches the performance of Aggregated-SP (see Figure 4). The difference in performance is notably high for Paintings domain, where specialized expertise is required to predict painting prices. Here, Partial-SP significantly outperforms common aggregation rules, showcasing its effectiveness in leveraging expert knowledge and correcting misinformation. For further details see Appendix G.3.

# 6 Simulated Model of Voter Behavior

In this section, we investigate whether there is any underlying probabilistic model that can explain the voters' behaviours when measured in terms of the vote and predictions. If successful, such a model will also enable us to theoretically analyze the sample complexity of SP algorithms (as we present in Section 7). In particular, we posit that a *concentric mixtures of Mallows model* can explain the users' reports (both vote and prediction) in the dataset. The concentric mixtures of Mallows model is a type of mixture models where there is one ground truth, but different groups of users have different dispersion parameters, and hence different distribution over observed preferences.

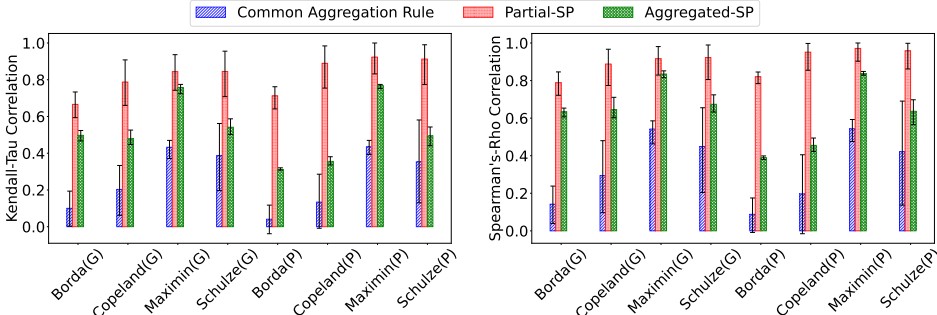

Figure 4: Comparing the predicted and ground-truth rankings for different aggregation rules using Kendall-Tau and Spearman's $\rho$ correlations (higher is better). The elicitation format is `Rank-Rank`; each comparison uses the same aggregation rule in the SP algorithm.

**Concentric mixtures of Mallows model.** We assume that each voter is likely to be an `expert` with probability $p(\ll 1)$ and a `non-expert` with probability $1 - p$. Given a ground truth ranking $\pi^\star$, an `expert` voter observes a ranking that is distributed according to a Mallows model with center $\pi^\star$ and dispersion parameter $\phi_E$. On the other hand, a `non-expert` voter observes a ranking that is again distributed according to a Mallows model with center $\pi^\star$, but with a larger dispersion parameter $\phi_{NE}$. In particular, the ranking observed by voter $i$ is distributed as

$$\Pr_s(\pi_i \mid \pi^\star) = p \cdot \Pr_s(\pi_i \mid \pi^\star, \phi_E) + (1 - p) \cdot \Pr_s(\pi_i \mid \pi^\star, \phi_{NE}) \tag{8}$$

where $\Pr_s(\pi \mid \pi^\star, \phi)$ is the standard Mallows model with dispersion $\phi$ i.e. $\Pr_s(\pi \mid \pi^\star, \phi) = \frac{\phi^{d(\pi, \pi^\star)}}{Z(\phi, m)}$. The term $Z(\phi, m)$ is the normalization constant, and is defined as $Z(\phi, m) = \sum_\pi \phi^{d(\pi, \pi^\star)}$. With a slight abuse of notation, we will write $Z(\phi)$ as $Z(\phi, m)$ since the number of alternatives in the ground truth is assumed to be fixed.

Note that, Equation (8) defines a distribution over complete preferences, but given a subset of size $k$ we can naturally extend this definition to define a distribution over partial preferences e.g. $\Pr_s(\sigma_i \mid \pi^\star)$ (eq. (4)), and posterior over partial preferences of other voters e.g. $\Pr_o(\sigma \mid \sigma_i)$ (eq. (6)).

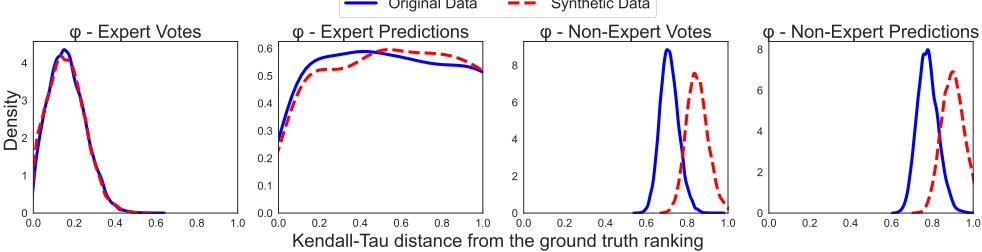

Figure 5: Comparison of inferred parameters of the *Concentric mixtures of Mallows model* for real data with all domains combined and synthetic data. The experts vote closer to and predict farther from the ground-truth. The non-experts vote and predict far from the ground truth. The proportion of experts in both datasets was found to be less than 20%.

We fit the mixture model eq. (8) on the real datasets and estimate the following parameters – proportion of experts ($p$), dispersion parameters of experts' votes ($\phi_{E\text{-}votes}$) and predictions ($\phi_{E\text{-}predictions}$), as well as the dispersion parameters of non-experts' votes ($\phi_{NE\text{-}votes}$) and predictions ($\phi_{NE\text{-}predictions}$). The parameters were inferred using Bayesian inference [24]. We also generated synthetic data using the concentric mixtures of Mallows model, and again used Bayesian inference to estimate the parameters. The details of estimation and data generation are provided in the appendix F. Figure 5 shows the posterior distributions for the dispersion parameters when the datasets of all the three domains are combined. We see that the synthetic data generation process accurately replicates real data characteristics, and highlights that the concentric mixtures of Mallows accurately model voters' behaviours on MTurk. Furthermore, Figure 6 also plots the same posterior distributions but only for

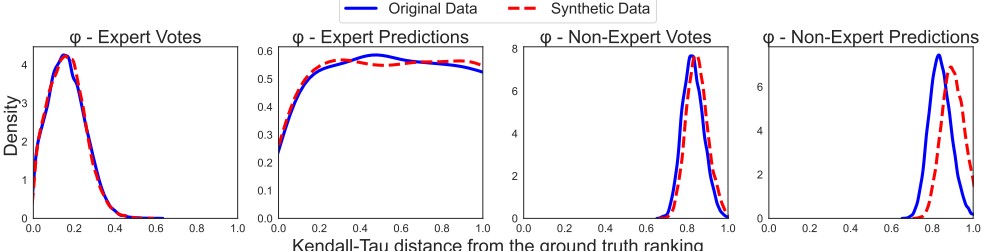

Figure 6: Comparison of inferred parameters of the *Concentric mixtures of Mallows model* for real data of Movie domain and synthetic data. The quality of model fit improves if the focus is on one single domain.

the Movies domain. Now we see almost perfect fit between the synthetic data and the original data. Further similarity in results between real and simulated data is described in Appendix G.6.

# 7 Analysis of Sample Complexity

In this section, we use a *concentric mixture of Mallows* models, and provide upper bound on the sample complexity of the surprisingly popular voting method with partial preferences. We start with a simple problem. Given a subset $T$ of size $k$, suppose our goal is to recover the true partial ranking over the alternatives in $T$, then how many samples does SP algorithm require?

We will analyze the following simplified version of the SP algorithm: Voter $i$ reports vote $\sigma_i$ over the subset $T$, which is used to build an estimate of $f(\sigma)$ for all $\sigma \in \Pi_s$. For each $\sigma$, the posterior report by the voter is a partial ranking drawn from the distribution $g(\cdot \mid \sigma)$. These reports are used to build an estimate $\widehat{g}(\sigma' \mid \sigma)$ for all $\sigma', \sigma$. Select partial ranking $\widehat{\sigma} \in \arg\max_\sigma \widehat{V}(\sigma) = \widehat{f}(\sigma) \cdot \sum_{\sigma \in \Pi_s} \frac{\widehat{g}(\sigma'|\sigma)}{\widehat{g}(\sigma|\sigma')}$.

It is impossible to recover the partial ground truth ranking if the fraction of the experts $p$ can be very small, or the dispersion parameter of the non-experts $\phi_{NE}$ can be very large. In order to ensure recovery of the true partial ranking we will make the following assumption.

**Assumption 1.** *The dispersion parameters $\phi_E, \phi_{NE}$, and the fraction of experts $p$ satisfy the following inequality,*

$$\left(\frac{p}{1-p}\right)^2 \geq 2 \cdot \left(\frac{Z(\phi_{NE})}{Z(\phi_E)}\right)^2 Z(\phi_{NE}, k)\phi_E^{k(k-1)/2}$$

*where $Z(\phi, k) = \sum_{\sigma:[k]\to[k]} \phi^{d(\sigma, \sigma^\star)}$.*

The next theorem states that sample complexity under the above assumption.

**Theorem 1.** *Suppose Assumption 1 holds, and the total number of samples $n \geq k!\sqrt{\frac{10k \log(2k/\delta)}{\mu}}$ where $\mu = p \cdot \frac{Z(\phi_E, m-k)}{Z(\phi_E)} \cdot \phi_E^{k(k-1)/2} + (1-p) \cdot \frac{Z(\phi_{NE}, m-k)}{Z(\phi_{NE})} \cdot \phi_{NE}^{k(k-1)/2}$. Then the surprisingly popular algorithm recovers true ranking over the subset $T$ of size $k$ with probability at least $1 - \delta$.*

Suppose $\phi_E \ll \phi_{NE} < 1$. Then Assumption 1 requires $\frac{p}{1-p} \geq \Omega\left(\phi_{NE}^{k^2/4+1}\phi_E^{k^2/4-1}\right)$, and it implies that if $\phi_{NE}$ is very large compared to $\phi_E$ (i.e. noisy non-experts) then we need a larger value of $p$ (i.e. more experts).

We provide the full proof of the theorem in Appendix I. The main ingredient of the proof is Lemma 2 which shows that under Assumption 1 there is a strict separation between the true prediction-normalized score of the true partial ranking and any other ranking. In fact, we show that $\overline{V}(\sigma^\star) \geq 2\overline{V}(\tau)$ for any $\tau$ with $d(\tau, \sigma^\star) \geq 1$. Given this result, we can apply standard concentration inequality to show that $\widehat{V}(\sigma)$ is close to $\overline{V}(\sigma)$ for all $\sigma$ when the number of samples is large, and $\widehat{V}(\sigma^\star)$ will be larger than $\widehat{V}(\tau)$ for any $\tau \neq \sigma^\star$. Therefore, picking the ranking with the largest empirical prediction-normalized score returns the correct ranking.

Note that the sample complexity grows proportional to $k!$ only because we compute prediction-normalized votes over all $k!$ partial rankings. If we are interested in recovering top $t$-alternatives then

it will grow proportional to $\binom{k}{t}$. Moreover, the subset size $k$ is assumed to be very small compared to the number of alternatives $m$, and Theorem 1 shows the benefit of applying SP algorithm to partial preferences. We can immediately apply Theorem 1 to a collection of subsets $S$ through a union bound, and extend our analysis to the Partial-SP algorithm. Let us assume that in the second stage of Partial-SP, we apply a $t$-*consistent* voting rule $f$ that recovers top-$t$ alternatives as long as each partial ranking in $S$ is correct.

**Corollary 1.** *Under the same setting as Theorem 1, suppose the number of samples from each subset in $S$ is $n \geq k!\sqrt{\frac{10k\log(2|S|k/\delta)}{\mu}}$. Then the Partial-SP algorithm with a $t$-consistent voting rule, recovers the top $t$ alternatives of the ground truth $\pi^\star$ with probability at least $1 - \delta$.*

Finally note that, the total sample complexity of $\tilde{O}(|S| \cdot k!\sqrt{k})$ is needed only because we adopt a naive version of the SP algorithm for proving theoretical guarantees. For the experiments, we adopt a pairwise version of the SP algorithm which applies SP-voting to each pair within a subset. We believe that under further assumptions, the total sample complexity can be reduced to $\tilde{O}(|S| \cdot k^2)$ with such a pairwise variant of the partial-SP algorithm, and we leave this analysis for the future.

# 8   Discussion and Future Work

We conclude by discussing some limitations and future directions. When dealing with partial preferences, even when majority have the correct information, effective preference elicitation or finding a necessary winner in most vote aggregation rules are often computationally intractable [11, 39, 18]. These challenges, together with the minority of experts, further highlight the efficacy of the SP approach in balancing information elicitation and accuracy, by employing additional prediction information. Future research can explore the setting of SP beyond the majority-minority dichotomy (e.g. informed, but not expert voters) or when malicious voters are present (e.g. in detecting misinformation). Theoretically, the sample complexity can be explored beyond Mallows model under other probabilistic models to enhance our understanding of this approach, particularly in notable applications such as political polling or collective moderation of online content.

## Acknowledgments and Disclosure of Funding

Hadi Hosseini acknowledges support from National Science Foundation (NSF) IIS grants #2144413 and #2107173. We thank the anonymous reviewers for their constructive feedback.

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

# Appendix

## Table of Contents

## A   Additional Related Work

**Information Aggregation.**    Information Aggregation by eliciting votes from voters is a well-studied problem in social choice theory. The aggregation rules proposed by De Condorcet [20], Borda [5], Copeland [16], Young [51], and many others focus on information aggregation by eliciting votes. These rules can be adapted to elicit ranked information from voters and then aggregate them into a single ranking representing the collective opinion of the crowd. [4]. Information Aggregation has also been examined from a statistical perspective, where the aggregated ranking is viewed as the maximum likelihood estimate of the population's rankings [20, 15, 50, 13]. Within this framework, individual votes are considered outcomes of probabilistic models such as the Thurstonian model, Bradley-Terry model, Mallows' model, or Plackett-Luce model [34].

**Partial Aggregation.** In situations where it is difficult or not necessary to elicit complete rankings from voters, partial preferences are used. Partial vote aggregation has different solution concepts [6]. Partial preferences can be used to conclude which alternatives are necessary and possible winners based on the preference profiles [26, 14, 47, 2, 3, 49]. Alternatively, a regret based approach can be used to assess the quality of a winning alternative where the optimal alternative is the one that minimizes regret [30]. Apart from these epistemic notions, a lot of work has been done on probabilistic analysis of winners from partial profiles [1, 23, 29, 31]. To minimize the information elicited from a population, it is crucial to understand the methods used for eliciting partial preferences. It can either be by minimizing the amount of information communicated by each voter in their answer [12, 45, 40] or by reducing the number of queries that each voter needs to answer [37, 14]. In either of these cases, the objective is still to determine the winner accurately. Cognitive complexity also plays an important part in this, as all these notions are highly correlated. Finally, the recovery of complete rankings from partial preferences is another solution concept that is studied [21, 41]. Due to the combinatorial nature of the rankings, winner determination, communication complexity, query complexity, and cognitive complexity are all relevant here. This is where our research work contributes.

**Surprisingly Popular Framework.** In their seminal work, Prelec [35], Prelec et al. [36] introduced the Surprisingly Popular (SP) algorithm, a novel second-order information based method that recovers truthful subjective data in scenarios where objective truth remains unknown. This framework has since been used to incentivize truthful behaviour in agents [35, 42, 43], mitigate biases in academic peer review [32], elicit expert knowledge [27], model thinking hierarchy of people without any prior [28], aggregate information [9] and recover ground-truth ranking [36, 25]. Our study builds upon this literature, specifically addressing the challenges in rank recovery. Originally, the SP algorithm by Prelec et al. [36] required data on all $m!$ potential rankings for $m$ alternatives , a requirement that becomes impractical as $m$ increases. Hosseini et al. [25] addressed this by developing a Surprisingly Popular Voting algorithm that leverages pairwise preference data across $\binom{m}{2}$ alternatives. However, this approach encountered scalability limitations when dealing with more than four alternatives in partial preference profiles. Our contribution lies in advancing this methodology by proposing a scalable generalization of the Surprisingly Popular Voting method for partial preferences, thus broadening its applicability and effectiveness.

## B    Formalism of Elicitation Formats

In this section, we formally define the elicitation formats used in our study. Let $v_i$ and $p_i$ denote the vote and prediction submitted by voter $i$. Let $T = \{a_1, a_2, \ldots, a_k\}$ denote the subset of alternatives of size $k$ that voters will report on and $\mathcal{L}(T)$ denote the set of all possible rankings of alternatives in $T$. Let $\sigma$ denote a ranking of the alternatives in $T$ and $\sigma(j)$ denote the alternative at the $j^{th}$ position in $\sigma$. The elicitation formats are defined as follows:

`Top-None`: Voter $i$ reports the top alternative in her observed noisy ranking, i.e., $v_i = \sigma(1)$, and does not provide any inference about other's aggregated votes.

`Top-Top`: Voter $i$ reports the top alternative in her observed noisy ranking, i.e., $v_i = \sigma(1)$, and provides the estimate of the most frequent alternative among the other voters, i.e., $p_i = \arg\max_{a \in T} \sum_{\sigma \in \mathcal{L}(T):\sigma(1)=a} \Pr_o(\sigma|\sigma_i)$.

`Top-Approval(3)`: Voter $i$ reports the top alternative in her observed noisy ranking, i.e., $v_i = \sigma(1)$, and provides the estimate of the top three most frequent alternatives, in no specific order, among the other voters., i.e., $p_i = \arg\max_{a,b,c \in T} \sum_{\sigma:\{a,b,c\} \subseteq \{\sigma(1),\sigma(2),\sigma(3)\}} \Pr_o(\sigma|\sigma_i)$.

`Top-Rank`: Voter $i$ reports the top alternative in her observed noisy ranking, i.e., $v_i = \sigma(1)$, and provides the estimate of other's rankings i.e $p_i \in \mathcal{L}(T)$ such that $\sum_{\sigma \in \mathcal{L}(\mathcal{A}):\sigma(1)=q_i(x)} \Pr_o(\sigma|\sigma_i) \geq \sum_{\sigma \in \mathcal{L}(T):\sigma(1)=q_i(y)} \Pr_o(\sigma|\sigma_i)$ for all $x > y$.

`Approval(2)-Approval(2)`: Voter $i$ reports the top two alternatives, in no specific order, in her observed noisy ranking, i.e., $v_i = \{\sigma(1),\sigma(2)\} = \{a,b\}$ with $a, b \in T$ in no particular order and provides the estimate of the top two most frequent alternatives, in no specific order, among the other voters., i.e., $p_i = \arg\max_{a,b \in T} \sum_{\sigma:\{a,b\} \subseteq \{\sigma(1),\sigma(2)\}} \Pr_o(\sigma|\sigma_i)$.

`Approval(3)-Rank:` Voter $i$ reports the top three alternatives, in no specific order, in her observed noisy ranking, i.e., $v_i = \{\sigma(1), \sigma(2), \sigma(3)\} = \{a, b, c\}$ with $a, b, c \in T$ in no particular order, and provides the estimate of other's rankings i.e, $p_i \in \mathcal{L}(T)$ such that $\sum_{\sigma \in \mathcal{L}(\mathcal{A}):\sigma(1)=q_i(x)} \Pr_o(\sigma|\sigma_i) \geq \sum_{\sigma \in \mathcal{L}(T):\sigma(1)=q_i(y)} \Pr_o(\sigma|\sigma_i)$ for all $x > y$.

`Rank-None:` Voter $i$ reports her entire observed noisy ranking, i.e., $v_i = \sigma_i$, and does not provide any inference about other's aggregated votes.

`Rank-Top:` Voter $i$ reports her entire observed noisy ranking, i.e., $v_i = \sigma_i$, and provides the estimate of the most frequent alternative among the other voters, i.e., $p_i = \arg\max_{a \in T} \sum_{\sigma \in \mathcal{L}(T):\sigma(1)=a} \Pr_o(\sigma|\sigma_i)$.

`Rank-Rank:` Voter $i$ reports her entire observed noisy ranking, i.e., $v_i = \sigma_i$, and provides the estimate of other's rankings i.e, $p_i \in \mathcal{L}(T)$ such that $\sum_{\sigma \in \mathcal{L}(\mathcal{A}):\sigma(1)=q_i(x)} \Pr_o(\sigma|\sigma_i) \geq \sum_{\sigma \in \mathcal{L}(T):\sigma(1)=q_i(y)} \Pr_o(\sigma|\sigma_i)$ for all $x > y$.

## C   Common Voting Rules

Vote aggregation rules are social choice functions that are used to aggregate individual votes to make conclusions about the collective opinion of a multi-candidate voting system [6]. Given below are the vote aggregation rules used in our study. We will only focus on aggregating ranked preferences.

| Number of Voters | Preference Profile |
|:---:|:---|
| 44 | $A \succ B \succ C \succ D$ |
| 24 | $B \succ C \succ D \succ A$ |
| 18 | $C \succ D \succ B \succ A$ |
| 14 | $D \succ C \succ B \succ A$ |

Table 1: Voter Preferences

### C.1   Borda

The Borda rule [5] is a voting rule in which voters order candidates by ranked preference, and candidates are awarded points based on their position in each voter's ranking. The winner is the candidate with the highest total score after all votes are counted. It can be mathematically represented as:

$$\text{Borda Score}(a) = \sum_{i=1}^{n} (m - 1 - \sigma_i^{-1}(a))$$

where $\sigma_i^{-1}(a)$ represents the position at which alternative $a$ is present. The aggregated ranking is derived by sorting the Borda scores of the alternatives in descending order.

**Example 1.** *Apply Borda Rule to the preference profile given in Table 1*

Table 1 provides the preference profiles of voters. Applying Borda Rule, we find that Borda Scores of A, B, C, D are 132, 192, 174, and 102 respectively. Thus arranging the scores in descending order results in the aggregated ranking of $B \succ C \succ A \succ D$.

### C.2   Copeland

The Copeland rule [16] is a voting method used to select a single winner from a set of candidates based on pairwise comparisons between each pair of candidates. In the Copeland method, each candidate receives a score based on the number of head-to-head contests they win against other candidates, with ties potentially receiving a half point for each candidate involved in the tie. It can be mathematically represented as:

$$\text{Copeland Score(a)} = \sum_{\substack{a,b \in A, \\ b \neq a}} \begin{cases} 1 & \text{if } V(a,b) > V(b,a) \\ 0.5 & \text{if } V(a,b) = V(b,a) \\ 0 & \text{if } V(a,b) < V(b,a) \end{cases}$$

where $V(a,b)$ represents all the voters who preferred $a$ over $b$. The aggregated ranking is derived by sorting the Copeland scores of the alternatives in descending order.

**Example 2.** *Apply Copeland Rule to the preference profile given in Table 1*

Applying Copeland Rule (with Borda tie-breaking) to the preference profiles given in Table 1 results in the following pairwise table:

| Pairwise Comparisons | Winner |
|:---:|:---:|
| A vs B | B |
| A vs C | C |
| A vs D | D |
| B vs C | B |
| B vs D | B |
| C vs D | C |

Table 2: Results of Pairwise Comparisons

Thus, arranging the alternatives in decreasing order of number of time they become winners results in the aggregated ranking of $B \succ C \succ D \succ A$.

### C.3 Maximin

The Maximin rule [51], also known as the Simpson-Kramer method , selects a winner from a set of candidates by considering the strength of a candidate's worst-case pairwise comparison against all other candidates. It identifies the candidate whose least favorable comparison is superior to those of the others, aiming to find the most robust candidate against the strongest opponent. This can be mathematically expressed as:

$$\text{Maximin Score(a)} = \min_{b \in A, b \neq a} V(a,b)$$

where $V(a,b)$ represents all the voters who preferred $a$ over $b$. The aggregated ranking is derived by sorting the Maximin scores of the alternatives in descending order.

**Example 3.** *Apply Maximin Rule to the preference profile given in Table 1*

In order to apply Maximin Rule to the preference profiles given in Table 1 we first analyze the worst pairwise defeat and its margin by the following table:

| Alternative | Worst Pairwise Defeat | Margin of Worst Pairwise Defeat |
|:---:|:---:|:---:|
| A | 56 | 12 |
| B | 0 | -12 |
| C | 68 | 36 |
| D | 86 | 72 |

Table 3: Analysis of Worst Pairwise Defeats

The alternative that has a higher score of worst pairwise defeat or that loses by a higher margin is considered worse off. Thus, arranging in ascending order of the scores of any of the column results in the aggregated ranking of $B \succ A \succ C \succ D$.

### C.4 Schulze

The Schulze rule [44] selects a ranking of a set of candidates based on the strength of preferences expressed by voters. The strength of a preference is considered to be the number of voters who prefer

one candidate over another. For every pair of candidates, a directed graph is constructed where the edges represent the strength of preference. The method then calculates the strongest path (defined as the weakest link in the path being as strong as possible) between every pair of candidates. A candidate wins if, for every other candidate, there exists a stronger (or equal strength) path to that candidate than from that candidate. It can be mathematically expressed as:

Initially, for all pairs of candidates $a, b$:

$$P(a,b) = \begin{cases} V(a,b) & \text{if } V(a,b) > V(b,a) \\ 0 & \text{otherwise} \end{cases}$$

Then, for each $a, b, c \in A$ with $a \neq b \neq c$, update:

$$P(a,b) = \max(P(a,b), \min(P(a,c), P(c,b)))$$

For each candidate $a$, calculate a comprehensive score that may involve the sum of all positive differences $P(a,b) - P(b,a)$ against other candidates $b$. The aggregated ranking is obtained by sorting these scores in descending order.

**Example 4.** *Apply Schulze Rule to the preference profile given in Table 1*

In order to apply Schulze Rule to the preference profile given in Table 1, we first generate a Directed Graph where the vertices denote the alternatives and the weight of the edges denote the score by which one alternative defeats the other. For the preferences in Table 1, we get the following graph:

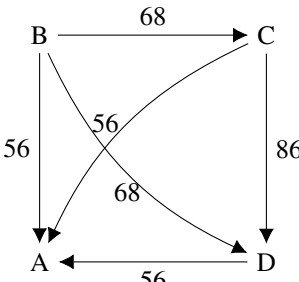

Figure 7: Directed Graph with Vertices Arranged as Corners of a Square

The Table 4 finds the strongest path for each pair of vertices:

| | A | B | C | D |
|---|---|---|---|---|
| A | X | 0 | 0 | 0 |
| B | B → 68 → C → 86 → D → 56 → A | X | B → 68 → C | B → 68 → C → 86 → D |
| C | C → 86 → D → 56 → A | 0 | X | C → 86 → D |
| D | D → 56 → A | 0 | 0 | X |

Table 4: Strongest Path between Vertices

Finally, the Table 4 provides the weakest link between each pair of vertices:

| | A | B | C | D |
|---|---|---|---|---|
| A | X | 0 | 0 | 0 |
| B | 56 | X | 68 | 68 |
| C | 56 | 0 | X | 86 |
| D | 56 | 0 | 0 | X |

Table 5: Strength of weakest link between vertices

Arranging the alternatives in decreasing order of their pairwise wins in Table 5 results in the aggregated ranking of $B \succ C \succ D \succ A$.

# D  Algorithms

Appendix D.1 details the approach used to extract information from various elicitation formats. Explanation and pseudocode for Partial-SP and Aggregated-SP is provided in Appendix D.2 and Appendix D.3, respectively.

## D.1  Extracting Reports from Voters

Algorithm 1 describes how information is extracted from different elicitation formats. The algorithm takes as input the votes ($\{v_i\}_{i\in[n]}$) and predictions ($\{p_i\}_{i\in[n]}$) of $n$ voters, pair $(a,b)$ of alternatives, and parameters $\alpha$ and $\beta$. Using grid search on the datasets from the three domains, it was observed that any $\alpha > 0.5$ and $\beta < 0.5$ can be used. In our experiments we use $\alpha = 0.55$ and $\beta = 0.1$. For votes and predictions expressed as top choices or approvals, if $a$ is the preferred alternative, $v_i^{(a,b)}$ is set to 1, with $p_i^{(a,b)}$ adjusted to $\alpha$ if the prediction aligns with the vote, or to $\beta$ otherwise; if $b$ is chosen, $v_i^{(a,b)}$ is 0, and $p_i^{(a,b)}$ is set to $1-\alpha$ or $1-\beta$, depending on prediction alignment. For ranks, $v_i^{(a,b)}$ indicates preference based on rank ordering, and $p_i^{(a,b)}$ reflects the confidence in this preference ($\alpha$ or $1-\alpha$ if aligned, $\beta$ or $1-\beta$ if not). The algorithm returns the processed vote and prediction reports.

---

**ALGORITHM 1:** Extract-Reports

**Input:** Votes $\{v_i\}_{i\in[n]}$, Predictions $\{p_i\}_{i\in[n]}$, pair $(a,b)$, and probabilities $\alpha > 0.5$, and $\beta < 0.5$.

**for** $i = 1, \ldots, n$ **do**

    /* Extract Vote Information                                                   */

    **if** $v_i$ *is elicited as rank* **then**

$$\text{Set } v_i^{(a,b)} = \begin{cases} 1 & \text{if } a \succ_{v_i} b \\ 0 & \text{o.w.} \end{cases}$$

    **else if** $v_i$ *is elicited as top or approval* **then**

$$\text{Set } v_i^{(a,b)} = \begin{cases} 1 & \text{if } v_i = a \\ 0 & \text{if } v_i = b \end{cases}$$

    **else**

        Ignore $(v_i, q_i)$

    /* Extract Prediction Information                                    */

    **if** $p_i$ *is elicited as rank* **then**

$$\text{Set } p_i^{(a,b)} = \begin{cases} \alpha & \text{if } a \succ_{p_i} b \text{ and } v_i^{(a,b)} = 1 \\ 1-\alpha & \text{if } b \succ_{p_i} a \text{ and } v_i^{(a,b)} = 1 \\ 1-\beta & \text{if } a \succ_{p_i} b \text{ and } v_i^{(a,b)} = 0 \\ \beta & \text{o.w.} \end{cases}$$

    **else if** $p_i$ *is elicited as top or approval* **then**

$$\text{Set } p_i^{(a,b)} = \begin{cases} \alpha & \text{if } p_i = a \text{ and } v_i^{(a,b)} = 1 \\ 1-\alpha & \text{if } p_i = b \text{ and } v_i^{(a,b)} = 1 \\ 1-\beta & \text{if } p_i = a \text{ and } v_i^{(a,b)} = 0 \\ \beta & \text{o.w.} \end{cases}$$

    **else**

        Set $p_i^{a,b} = \frac{1}{2}$

**return** $\left( \{ v_i^{(a,b)}, p_i^{(a,b)} \}_{i\in[n]} \right)$

---

## D.2  Partial-SP

Algorithm 2 describes the proposed Partial-SP aggregation approach. The algorithm takes as input the number of voters ($n$), number of alternatives ($m$), set of all subsets that voters voted on ($S$), voters' votes ($\{v_{i,j}\}_{i\in[n],j\in S}$), voters' predictions ($\{p_{i,j}\}_{i\in[n],j\in S}$), parameters $\alpha$ and $\beta$ representing the conditional probabilities that would be returned for the predictions, and Vote Aggregation Rule ($\mathcal{V}$). For $n$ voters and $m$ alternatives, depending on the elicitation format, the voters will be providing votes ($v_{i,j}$) and predictions ($p_{i,j}$) as Top choices, Approvals($t$), or Rankings over a subset $S_j \in S$. Additionally, we use Borda, Copeland, and Maximin aggregation rule for $\mathcal{V}$.

For every pair of alternatives $(a,b)$ within a subset $S_j$, we extract information about the number of people that voted for $a \succ b$ and $b \succ a$ represented by $f(a \succ b)$ and $f(b \succ a)$ respectively, and the conditional probability of their predictions ($g(0|0), g(0|1), g(1|0), g(1|1)$). Refer to Appendix D.1 for a detailed explanation of how information is extracted for different elicitation formats. With this, the prediction normalized score $\bar{V}(a \succ b)$ and $\bar{V}(b \succ a)$ is calculated and a higher prediction

normalized score decides the correct ordering for each pair $(a, b)$ within a subset $S_j$. This results in a partial ground-truth ranking representing the correct relative ordering for the alternatives within that subset. $Q$ represents the set of partial ground-truth ranking for all subsets. Finally, Vote-Aggregation rule $\mathcal{V}$ is applied on $Q$ to find the complete ground-truth ordering of the $m$ alternatives.

---

**ALGORITHM 2:** Partial-SP

---

**Input:** Number of Voters $n$, Subsets of alternatives $S$, Votes $\{v_{i,j}\}_{i \in [n], j \in S}$, Predictions $\{p_{i,j}\}_{i \in [n], j \in S}$, probabilities $\alpha > 0.5$ and $\beta < 0.5$, and Vote-Aggregation rule $\mathcal{V}$

$Q \leftarrow \phi$

**for** $j = 1, \ldots, |S|$ **do**

  $G \leftarrow \phi$

  /* Apply SP-voting on votes and predictions for each subset         */

  **for** *each pair of alternatives* $(a, b)$ *in* $S_j$ **do**

    $\left(\{v_i^{(a,b)}, p_i^{(a,b)}\}_{i \in [n]}\right) \leftarrow$ Extract-Reports$(\{v_{ij}, p_{ij}\}_{i \in [n], j \in S}, pair(a, b), \alpha, \beta)$

    /* Signal 1 (resp. 0) corresponds to $a \succ b$ (resp. $b \succ a$).     */

    $N_{a \succ b} = \left\{c : v_c^{(a,b)} = 1\right\}$

    $N_{b \succ a} = \left\{c : v_c^{(a,b)} = 0\right\}$

    $f(a \succ b) = \sum_i 1\left\{v_i^{(a,b)} = 1\right\} / (|N_{a \succ b}| + |N_{b \succ a}|)$

    $f(b \succ a) = 1 - f(a \succ b)$

    $g(1 \mid 1) = \frac{1}{|N_{a \succ b}|} \sum_{i \in N_{a \succ b}} p_i$ and $P(0 \mid 1) = 1 - P(1 \mid 1)$

    $g(1 \mid 0) = \frac{1}{|N_{b \succ a}|} \sum_{i \in N_{b \succ a}} p_i$ and $P(0 \mid 0) = 1 - P(1 \mid 0)$

    /* Compute prediction-normalized vote         */

    $\bar{V}(a \succ b) = f(a \succ b) \sum_i \dfrac{g\left(v_i^{(a,b)} \mid 1\right)}{g\left(1 \mid v_i^{(a,b)}\right)}$

    $\bar{V}(b \succ a) = f(b \succ a) \sum_i \dfrac{g\left(v_i^{(a,b)} \mid 0\right)}{g\left(0 \mid v_i^{(a,b)}\right)}$

    /* Ties are broken uniformly at random         */

    **if** $\bar{V}(a \succ b) < \bar{V}(b \succ a)$ **then**

      $G \leftarrow G \cup a \succ b$

    **else**

      $G \leftarrow G \cup b \succ a$

  $Q_j \leftarrow Q_j \cup G$

$GT \leftarrow \mathcal{V}(Q)$

**return** $GT$

---

### D.3 Aggregated-SP

Algorithm 3 describes the proposed Aggregated-SP aggregation approach. The algorithm takes as input the number of voters ($n$), number of alternatives ($m$), set of all subsets that voters voted on ($S$), voters' votes ($\{v_{i,j}\}_{i \in [n], j \in S}$), voters' predictions ($\{p_{i,j}\}_{i \in [n], j \in S}$), parameters $\alpha$ and $\beta$ representing the conditional probabilities that would be returned for the predictions, and Vote Aggregation Rule ($\mathcal{V}$). For $n$ voters and $m$ alternatives, depending on the elicitation format, the voters will be providing votes ($v_{i,j}$) and predictions ($p_{i,j}$) as Top choices, Approvals($t$), or Rankings over a subset $S_j \in S$. Additionally, we use Borda, Copeland, and Maximin aggregation rule for $\mathcal{V}$.

For every subset $S_j$, we aggregate the votes using $\mathcal{V}$, resulting in the set of partial aggregated subsets represented by $Q$. $Q$ is a dictionary containing the alternative and its corresponding score according to the aggregation rule $\mathcal{V}$. We now apply SP-Algorithm on $Q$. For every pair of alternatives $(a, b)$ within a partial aggregated subset $Q_j$, we extract information about the conditional probability of their predictions $(g(0|0), g(0|1), g(1|0), g(1|1))$. Refer to Appendix D.1 for a detailed explanation of how information is extracted for different elicitation formats. With this, the prediction normalized score $\bar{V}(a \succ b)$ and $\bar{V}(b \succ a)$ is calculated where we use the scores of the alternatives represented by $Q(a)$ and $Q(b)$. A higher prediction normalized score decides the correct ordering for each pair $(a, b)$. Parsing all pairs, results in the complete ground-truth ordering of the $m$ alternatives.

**ALGORITHM 3:** Aggregated-SP Aggregation

**Input:** Number of Voters $n$, Subsets of alternatives $S$, Votes $\{v_{i,j}\}_{i\in[n],j\in S}$, Predictions $\{p_{i,j}\}_{i\in[n],j\in S}$,
probabilities $\alpha > 0.5$ and $\beta < 0.5$, and Vote-Aggregation rule $\mathcal{V}$

$Q \leftarrow \phi$
$GT \leftarrow \phi$
**for** $j = 1, \ldots, |S|$ **do**
$\quad G \leftarrow \phi$
$\quad G \leftarrow \mathcal{V}(v_{i,j})$

$\quad Q_j \leftarrow Q_j \cup G$
$\quad$ /* Apply pairwise SP-voting on aggregated votes and non-aggregated
$\quad$ predictions                                                                */
$\quad$ **for** *each pair of alternatives* $(a, b)$ *in* $Q_j$ **do**
$\quad\quad \left( \{v_i^{(a,b)}, p_i^{(a,b)}\}_{i\in[n]} \right) \leftarrow$ Extract-Reports$(\{v_{ij}, p_{ij}\}_{i\in[n],j\in S}, pair(a,b), \alpha, \beta)$
$\quad\quad N_{a \succ b} = \left\{ c : v_c^{(a,b)} = 1 \right\}$
$\quad\quad N_{b \succ a} = \left\{ c : v_c^{(a,b)} = 0 \right\}$
$\quad\quad g(1 \mid 1) = \frac{1}{|N_{a \succ b}|} \sum_{i \in N_{a \succ b}} p_i$ and $P(0 \mid 1) = 1 - P(1 \mid 1)$
$\quad\quad g(1 \mid 0) = \frac{1}{|N_{b \succ a}|} \sum_{i \in N_{b \succ a}} p_i$ and $P(0 \mid 0) = 1 - P(1 \mid 0)$
$\quad\quad$ /* Compute prediction-normalized vote                                      */

$\quad\quad \bar{V}(a \succ b) = Q(a) \sum_i \dfrac{g\left(v_i^{(a,b)} | 1\right)}{g\left(1 | v_i^{(a,b)}\right)}$

$\quad\quad \bar{V}(b \succ a) = Q(b) \sum_i \dfrac{g\left(v_i^{(a,b)} | 0\right)}{g\left(0 | v_i^{(a,b)}\right)}$

$\quad\quad$ /* Ties are broken uniformly at random                                      */
$\quad\quad$ **if** $\bar{V}(a \succ b) < \bar{V}(b \succ a)$ **then**
$\quad\quad\quad |$ $GT \leftarrow GT \cup a \succ b$
$\quad\quad$ **else**
$\quad\quad\quad |$ $GT \leftarrow GT \cup b \succ a$
**return** $GT$

# E    Additional Details of Experimental Design

This section provides additional details about the MTurk study

**Tutorial.** Prior to engaging with each set of 6 questions within a specific elicitation format, participants completed a tutorial designed to evaluate their understanding of the voting process and prediction tasks. To proceed, participants had to accurately apply these beliefs within the voting and prediction framework, ensuring they were adequately prepared.

**Reviews.** Following the completion of each set of 6 questions, participants were asked to evaluate the preceding questions' elicitation format in terms of difficulty (ranging from "Very Easy" to "Very Difficult") and expressiveness (from "Very Little" to "Very Significant"). Although question complexity was standardized within each domain, the domains themselves varied considerably in difficulty. To mitigate potential bias from implicit comparisons between the two elicitation formats assigned to each participant, the sequence of domains in the first set of questions was mirrored in the subsequent set. This methodological approach ensured consistency and fairness in the evaluation of the elicitation formats, thereby enhancing the reliability of participants' feedback

**Response qualifications & payment.** To ensure reliable responses, we established several qualification criteria for participants in our study on MTurk. Participants were required to have: (a) a minimum approval rate of 90% for their previous tasks, (b) completed at least 100 tasks on the platform, and (c) specified the region as US and Canada (to ensure language proficiency). To check attentiveness of the participants, we included an additional quiz that repeated one of the previous questions. The compensation structure included a base payment of 50 cents for completing the survey, which encompassed tutorials, questions, and evaluations. Additionally, a 50-cent bonus was offered for accurately completing the attentiveness quiz.

## F  Additional Details of Simulation

### F.1  Parameter Inference for the concentric mixtures of Mallows model

To assess the accuracy of the simulations, we fit the concentric mixtures of Mallows model to both the real and simulated data to infer the model parameters. This process allows us to compare the inferred parameters, thereby evaluating how effectively the model captures the underlying patterns in the data. Specifically, we infer the proportion of experts ($p$), dispersion parameters of experts' votes ($\phi_{E\text{-}votes}$) and predictions ($\phi_{E\text{-}predictions}$), as well as the dispersion parameters of non-experts' votes ($\phi_{NE\text{-}votes}$) and predictions ($\phi_{NE\text{-}predictions}$). For the real data, the distribution of these parameters can help us in understanding the voting behavior of experts and non-experts. For the synthetic data, generated based on known parameters, we can check if the model accurately recovers these parameters.

The parameters are inferred using Bayesian inference, implemented through the No-U-Turn Sampler (NUTS) [24], an extension of Hamiltonian Monte Carlo (HMC) available in Stan [8]. Before sampling, we calculate the Kendall-Tau distance between the ground-truth ranking and the votes ($\tau_{votes}$) and predictions ($\tau_{predictions}$) of all the voters. We then define the priors for our parameters as follows:

$$p \sim \beta(1, 2.5)$$
$$\phi_{E\text{-}votes} \sim \mathcal{N}(0.15, 0.075)$$
$$\phi_{E\text{-}predictions} \sim \mathcal{N}(0.7, 0.3)$$
$$\phi_{NE\text{-}votes} \sim \mathcal{N}(0.7, 0.3)$$
$$\phi_{NE\text{-}predictions} \sim \mathcal{N}(0.7, 0.3)$$

We then combine the likelihood of the parameters of our mixture model and perform inference over the following target function:

$$
\begin{aligned}
\text{target}+ = \log \text{mix} \,(p, \\
\mathcal{N}(\tau_{\text{votes}}[n] \mid 0, \phi_{E\text{-}votes}) + \mathcal{N}(\tau_{\text{predictions}}[n] \mid 0, \phi_{E\text{-}predictions}), \\
\mathcal{N}(\tau_{\text{votes}}[n] \mid 0, \phi_{NE\text{-}votes}) + \mathcal{N}(\tau_{\text{predictions}}[n] \mid 0, \phi_{NE\text{-}predictions}))
\end{aligned}
$$

The log-likelihood function incorporates the observed $\tau_{votes}$ and $\tau_{predictions}$ using the mixture model to account for the possibility that each voter could be an expert or a non-expert. We run four chains, each for 4000 iterations with 1000 iterations of warm-up for the NUTS algorithm. The algorithm explores the parameter space and updates the parameter estimates iteratively based on the input data and priors.

### F.2  Synthetic Data Generation

The synthetic data was generated by simulating voter behavior using the concentric mixtures of Mallows model to construct preference rankings. This approach allows for the simulation of both expert and non-expert voters, with experts' votes closely aligning with a ground truth ranking and non-experts showing a broader dispersion in their preferences. The subsets to be voted on were generated as described in Section 4.

**Voter Classification.** To simulate the voting process effectively, voters are initially classified into experts and non-experts. Since we need the experts to be in the minority, we determine the probability of a voter being an expert by sampling the proportion parameter as $p \sim \beta(1, 2.5)$.

**Voting Simulation**: The voting behavior is simulated by the concentric mixtures of Mallows model.

$$\text{Pr}_s(\pi_i \mid \pi^\star) = p \cdot \text{Pr}_s(\pi_i \mid \pi^\star, \phi_{\text{E}}) + (1 - p) \cdot \text{Pr}_s(\pi_i \mid \pi^\star, \phi_{\text{NE}}) \tag{9}$$

where $\phi_{\text{E}} \sim N(0.15, 0.075)$ and $\phi_{\text{NE}} \sim N(0.9, 0.4)$. Kendall-Tau distance is used as the distance metric between the rankings $\pi_i$ and $\pi^\star$.

# G  Missing Results and Analysis

## G.1  Evaluation Metrics

To quantitatively assess the alignment between the rankings derived from the voting rule, denoted as $\sigma'$, and the ground truth ranking, $\sigma^*$, we use metrics described in the following subsections.

### G.1.1  Pairwise hit rate

This metric evaluates the accuracy of the voting rule in identifying the correct relative order between pairs of alternatives, focusing on pairs with an increasing distance in their positions in the ground truth ranking:

$$\text{Pairwise hit rate} = \frac{1}{|P|} \sum_{(i,j) \in P} \mathbf{1}((\sigma'(i) < \sigma'(j)) = (\sigma^*(i) < \sigma^*(j)))$$

where $P$ represents the set of pairs determined by the difference in their positions in $\sigma^*$, and $\mathbf{1}$ is the indicator function.

### G.1.2  Top-$t$ hit rate

The Top-$t$ hit rate metric quantitatively assesses the accuracy of a ranking algorithm by measuring the presence of the top-$t$ elements from the ground truth ranking within the top-$t$ elements of the predicted ranking. The formula for calculating the Top-$t$ hit rate for rankings up to a given $t$ is given by:

$$\text{Top-}t \text{ hit rate} = \frac{|\text{Top-}t \text{ elements in ground truth} \cap \text{Top-}t \text{ elements in predicted ranking}|}{t}$$

### G.1.3  Kendall-Tau Correlation Coefficient

The Kendall-Tau correlation coefficient is a measure of the ordinal association between two rankings:

$$\tau(\sigma', \sigma^*) = \frac{2}{n(n-1)} \sum_{i<j} \mathbf{1}((\sigma'(i) - \sigma'(j))(\sigma^*(i) - \sigma^*(j)) > 0) - 1$$

where $n$ is the number of elements in the ranking.

### G.1.4  Spearman's $\rho$

The Spearman's $\rho$ or Spearman correlation coefficient between $\sigma'$ and $\sigma^*$ quantifies the rank correlation:

$$\rho(\sigma', \sigma^*) = 1 - \frac{6 \sum_{i=1}^{n} d_i^2}{n(n^2 - 1)}$$

with $d_i = \sigma'(i) - \sigma^*(i)$ representing the rank difference of each element $i$ between $\sigma'$ and $\sigma^*$.

**Bootstrapping.**  To ensure the robustness of our estimates, we used bootstrapping to approximate the sampling distribution of various statistics. This method offers insights into the variance and bias of our estimates without relying on the stringent assumptions required by traditional parametric methods. Bootstrapping is particularly advantageous when the distribution of metric values is unknown or does not conform to common distributional assumptions. After generating the bootstrapped distribution of metric values, we calculated the 95% confidence interval for each metric. Reporting these intervals alongside the metric values serves two purposes: it quantifies the uncertainty of our estimates, providing a transparent measure of their statistical precision, and it enhances the credibility of our findings by acknowledging the variability and potential error margins associated with our estimates.

## G.2  Response-Time, Difficulty and Expressiveness

Here, we measure the response time, difficulty, and expressiveness of the elicitation formats we used in our study.

- Response Time: We measure the average response time spent by the participants on each question for each elicitation format in our survey. This measure gives us an idea of the cognitive load perceived for each elicitation format [38].

- Perceived Difficulty: In the review phase of our MTurk survey, we asked participants to select from 'Very Easy', 'Easy', 'Neutral', 'Difficult', and 'Very Difficult' to subjectively indicate the ease of answering questions within each elicitation format. This approach provides a measure of the perceived difficulty associated with different elicitation formats.

- Perceived Expressiveness: In the review phase of our MTurk survey, we prompted participants to select from the options 'Very Little', 'Little', 'Adequate', 'Significant', and 'Very Significant' to indicate the amount of information they could convey using each elicitation format.

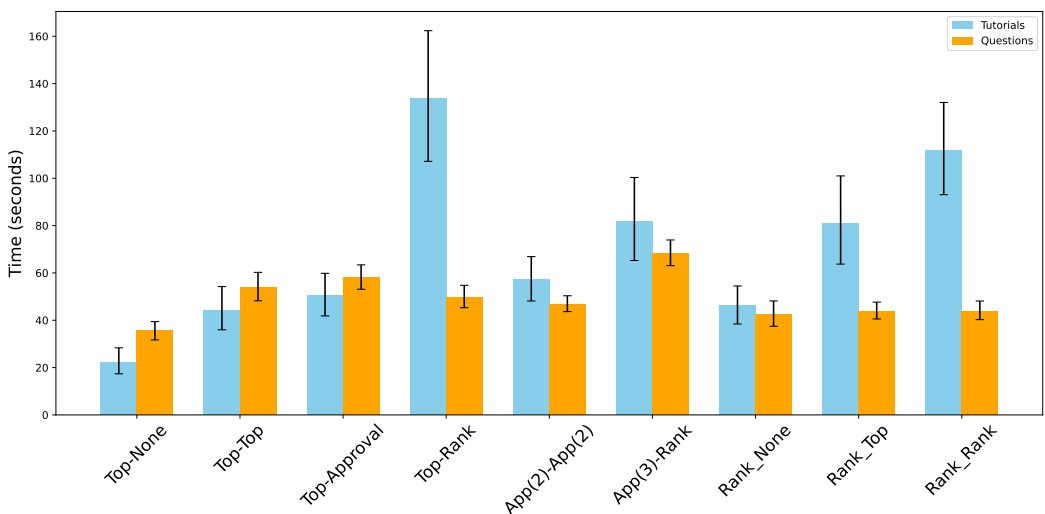

Figure 8: The figure shows the average time spent by the participants on each tutorial and question for different elicitation formats across all domains. As expected, the participants spent similar or more time on the tutorial than on the questions. Additionally, the only elicitation format that has a statistical significance for the Questions is `Approval(3)-Rank`, where more time is spent by the participants. Thus, for all other elicitation formats, the participants face a similar cognitive complexity while responding to the questions.

### G.3    Missing Figures for Predicting the Ground-Truth Ranking

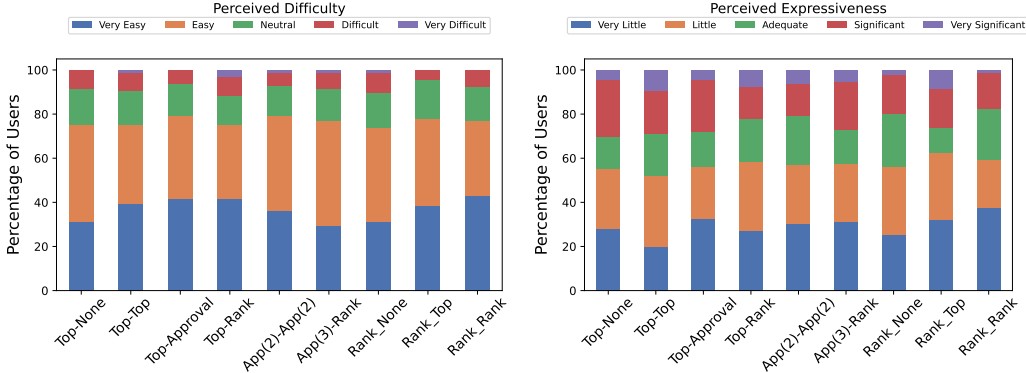

Figure 9: The figure shows the difficulty (easier is better) and expressiveness (higher is better) of different elicitation formats as reported by the participants. A higher percentage of participants found the tasks to be relatively easy, indicating that they could answer questions effortlessly across all elicitation formats. Conversely, they demonstrated similar expressiveness not strongly leaning to either side of the scale.

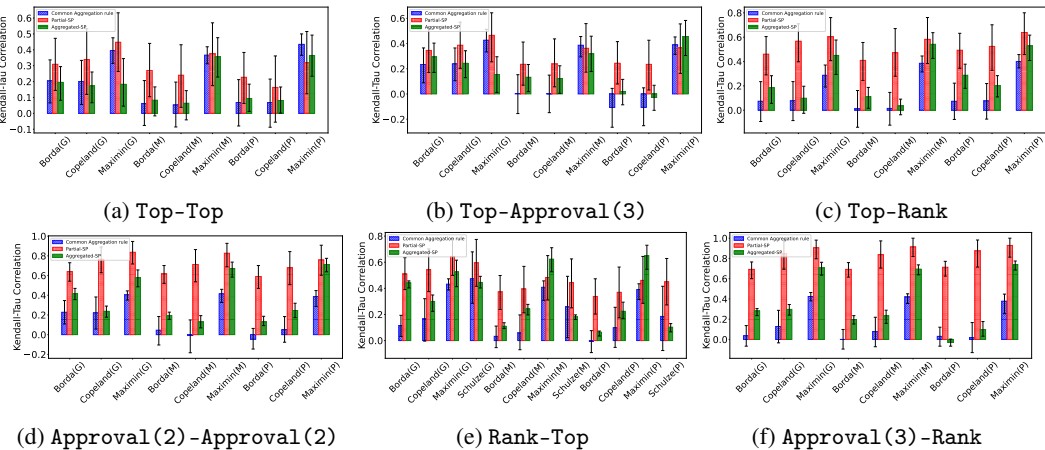

Figure 10: The plots show the Kendall-Tau Correlation between rankings derived from Common Aggregation Rules (blue), Partial-SP(red), and Aggregated-SP(red) for `Top-Top`, `Top-Approval(3)`, `Top-Rank`, `Approval(2)-Approval(2)`, `Rank-Top`, and `Approval(3)-Rank` elicitation formats across Geography(G), Movies(M), and Paintings(P) domains. A high Kendall-Tau Correlation implies higher pairwise alignment of alternatives between the ground-truth ranking and the aggregated ranking. The usage of different aggregation rules for Partial-SP and Aggregated-SP has similar impact on the outcome. However, the performance improves with an increase in information elicited as seen by the high correlation and increases statistical difference between the conventional methods and proposed methods. For example, `Approval(2)-Approval(2)` recovers ground-truth ranking more accurately than `Top-Top`. Note: We see Schulze method in `Rank-Top` as it only works for preference data.

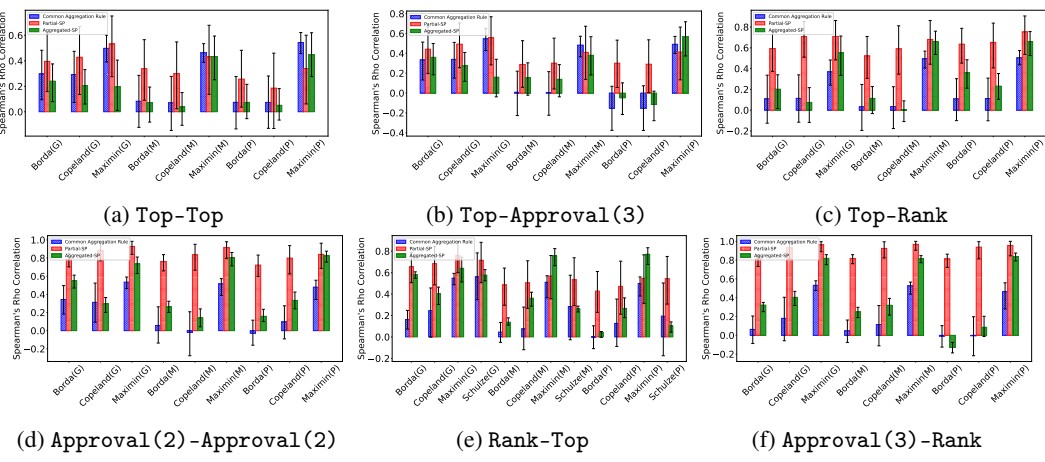

Figure 11: The plots show the Spearman's $\rho$ Correlation between rankings derived from Common Aggregation Rules (blue), Partial-SP(red), and Aggregated-SP(red) for `Top-Top`, `Top-Approval(3)`, `Top-Rank`, `Approval(2)-Approval(2)`, `Rank-Top`, and `Approval(3)-Rank` elicitation formats across Geography(G), Movies(M), and Paintings(P) domains. A high Spearman's $\rho$ Correlation implies higher alignment between the ground-truth ranking and the aggregated ranking. The usage of Maximin aggregation rule for Partial-SP and Aggregated-SP has a better impact on the outcome as compared to other common aggregation rules. Additionally, the performance improves with an increase in information elicited as seen by the high correlation and increases statistical difference between the conventional methods and proposed methods. For example, `Approval(3)-Rank` recovers ground-truth ranking more accurately than `Top-Approval(3)`. Note: We see Schulze method in `Rank-Top` as it only works for preference data.

## G.4 Missing Figures for Partial-SP

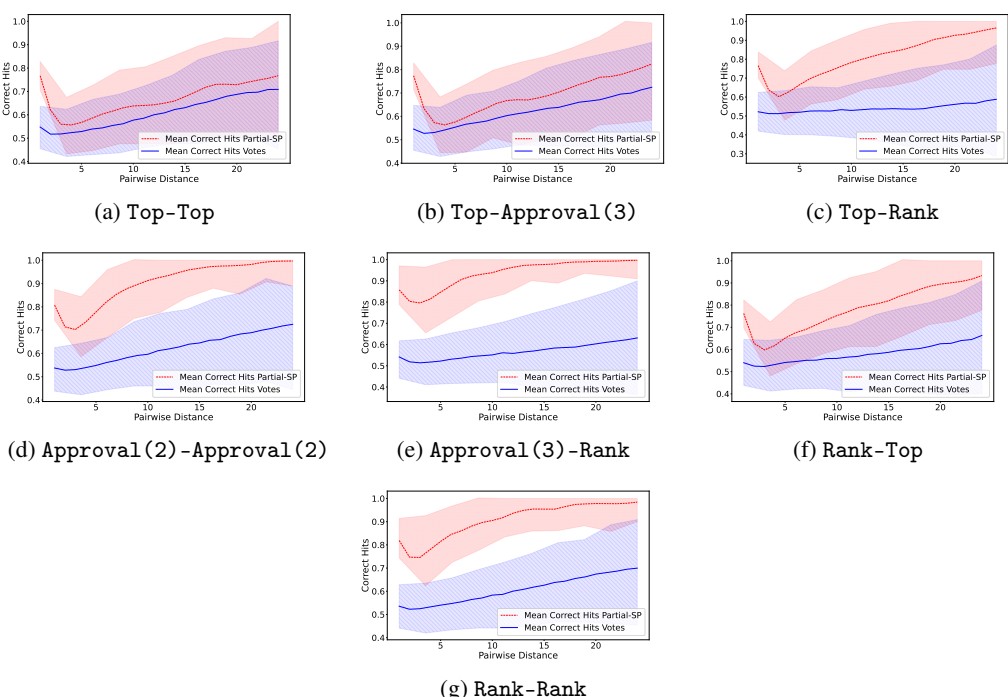

Figure 12: The figures show the effect of different elicitation formats for ground-truth recovery using Copeland and Partial-SP using metric defined in Section G.1.1. The metric assesses the number of pairs that were correctly predicted according to the aggregation rule based on their increasing distance in the ground-truth ranking. Comparable performance between `Approval(2)-Approval(2)`, `Approval(3)-Rank`, and `Rank-Rank` show that eliciting Approvals on half the size of the subset recovers truth as good as eliciting Ranking over the subset.

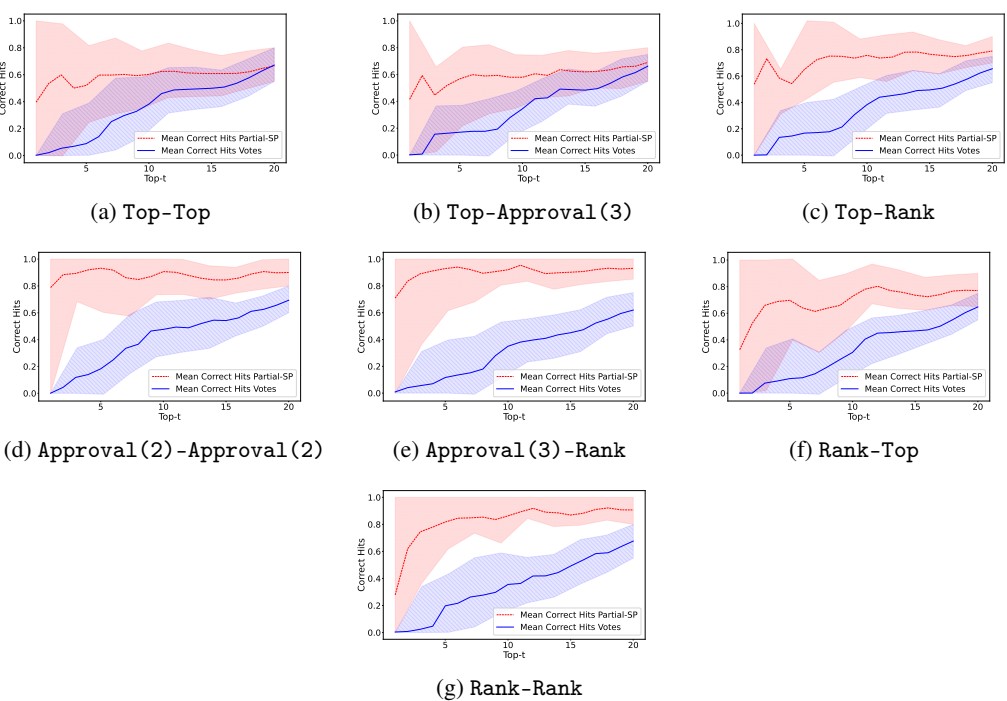

(a) `Top-Top`
(b) `Top-Approval(3)`
(c) `Top-Rank`
(d) `Approval(2)-Approval(2)`
(e) `Approval(3)-Rank`
(f) `Rank-Top`
(g) `Rank-Rank`

Figure 13: The figures show the effect of different elicitation formats for ground-truth recovery using Copeland and Partial-SP using metric defined in Section G.1.2. We again observe comparable performance between `Approval(2)-Approval(2)`, `Approval(3)-Rank`, and `Rank-Rank` show that eliciting Approvals on half the size of the subset recovers truth as good as eliciting Ranking over the subset.

## G.5 Missing Figures for Aggregated-SP

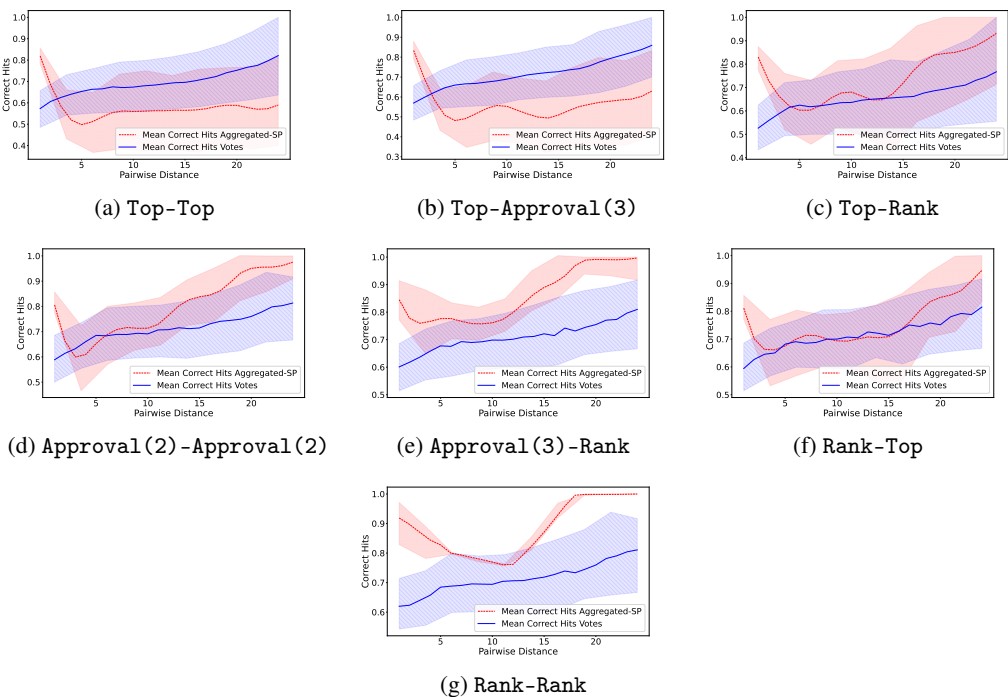

(a) `Top-Top`

(b) `Top-Approval(3)`

(c) `Top-Rank`

(d) `Approval(2)-Approval(2)`

(e) `Approval(3)-Rank`

(f) `Rank-Top`

(g) `Rank-Rank`

Figure 14: The figures show the effect of different elicitation formats for ground-truth recovery using Maximin and Aggregated-SP using metric defined in Section G.1.1. Improved performance in `Approval(3)-Rank`, and `Rank-Rank` are consistent with the observations made in Figure 12 except for `Approval(2)-Approval(2)` where no statistical significance is observed.

## G.6 Comparing Performance between Real and Simulated Data

Figure 16 shows the performance of Partial-SP with Copeland Aggregation on Real Data and Simulated Data using metrics described in Section G.1.1 and G.1.2. In both of the metrics, we observe similar trends across various pairwise distances, and top-t metrics.

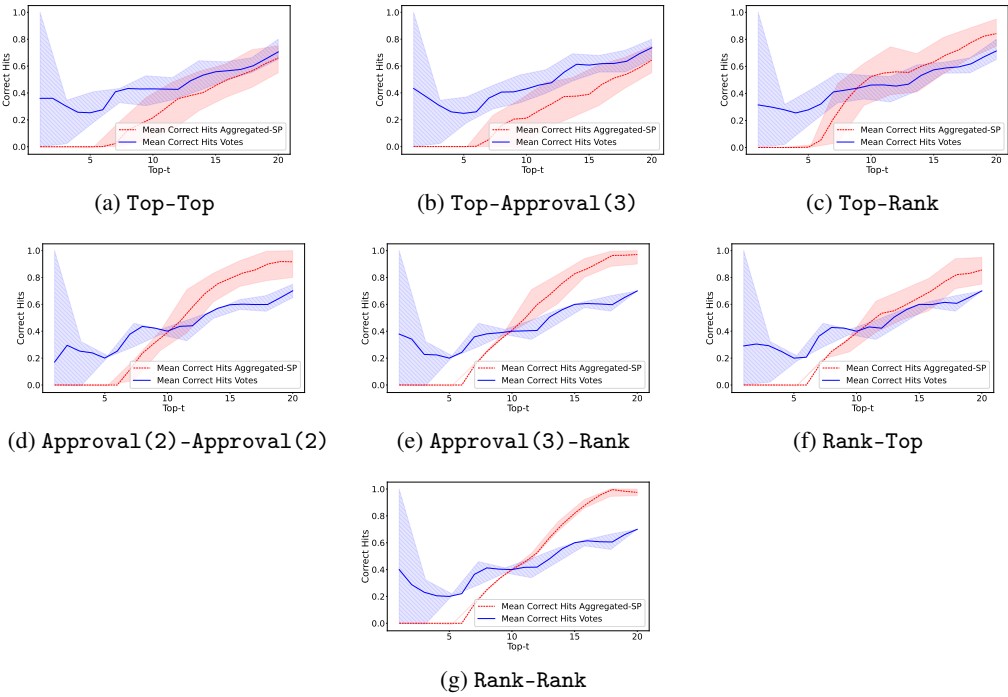

(a) `Top-Top`  (b) `Top-Approval(3)`  (c) `Top-Rank`

(d) `Approval(2)-Approval(2)`  (e) `Approval(3)-Rank`  (f) `Rank-Top`

(g) `Rank-Rank`

Figure 15: The figures show the effect of different elicitation formats for ground-truth recovery using Maximin and Aggregated-SP using metric defined in Appendix G.1.2. Improved performance, especially after $t = 10$, in `Approval(2)-Approval(2)`, `Approval(3)-Rank`, and `Rank-Rank` are consistent with the observations made in Figure 13.

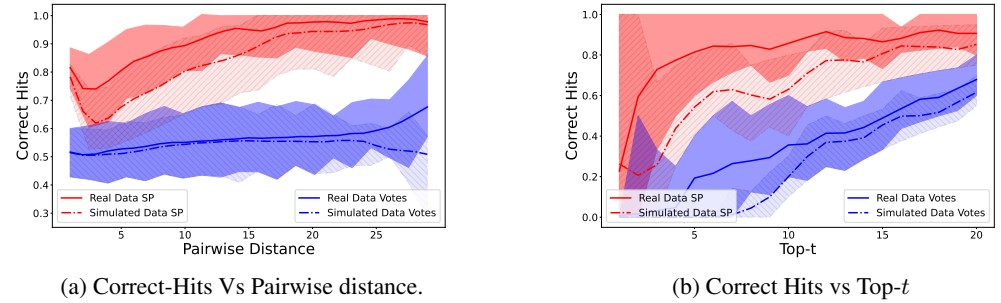

(a) Correct-Hits Vs Pairwise distance.  (b) Correct Hits vs Top-$t$

Figure 16: Comparison of pairwise and Top-$t$ hit rate for Copeland-Aggregated Partial-SP and Copeland rule for `Rank-Rank` on Real and Simulated Data. Similar trends are noticed in real and simulated data.

# H  Missing Proofs

## H.1   Proof of Theorem (1)

*Proof.* We first establish a lower bound on $g(\sigma \mid \sigma')$ for any $\sigma, \sigma'$.

$$g(\sigma \mid \sigma') = \sum_{\tilde{\sigma}} \mathrm{Pr}_g(\tilde{\sigma} \mid \sigma') \mathrm{Pr}_s(\sigma \mid \tilde{\sigma}) \geq \min_{\tilde{\sigma}} \mathrm{Pr}_s(\sigma \mid \tilde{\sigma}) \sum_{\tilde{\sigma}} \mathrm{Pr}_g(\tilde{\sigma} \mid \sigma') = \min_{\tilde{\sigma}} \mathrm{Pr}_s(\sigma \mid \tilde{\sigma})$$

Under the assumption of uniform prior we have,

$$
\begin{aligned}
\mathrm{Pr}_s(\sigma \mid \tilde{\sigma}) &= \sum_{\pi:\pi \triangleright \tilde{\sigma}} \frac{\mathrm{Pr}(\pi)}{\mathrm{Pr}(\tilde{\sigma})} \mathrm{Pr}_s(\sigma \mid \pi) \\
&= \frac{1}{(m-k)!} \sum_{\pi:\pi \triangleright \tilde{\sigma}} \mathrm{Pr}_s(\sigma^\star \mid \pi) \\
&= \frac{1}{(m-k)!} \sum_{\pi:\pi \triangleright \tilde{\sigma}} \sum_{\pi':\pi' \triangleright \sigma} \mathrm{Pr}_s(\pi' \mid \pi) \\
&= \frac{1}{(m-k)!} \sum_{\pi:\pi \triangleright \tilde{\sigma}} \sum_{\pi':\pi' \triangleright \sigma} p \cdot \frac{\phi_E^{d(\pi',\pi)}}{Z(\phi_E)} + (1-p) \cdot \frac{\phi_{NE}^{d(\pi',\pi)}}{Z(\phi_{NE})} \\
&= \frac{1}{(m-k)!} \sum_{\pi:\pi \triangleright \tilde{\sigma}} \left( p \cdot \frac{Z(\phi_E, m-k)}{Z(\phi_E)} \cdot \phi_E^{d(\sigma,\tilde{\sigma})} + (1-p) \cdot \frac{Z(\phi_{NE}, m-k)}{Z(\phi_{NE})} \cdot \phi_{NE}^{d(\sigma,\tilde{\sigma})} \right) \\
&= \underbrace{p \cdot \frac{Z(\phi_E, m-k)}{Z(\phi_E)} \cdot \phi_E^{d(\sigma,\tilde{\sigma})} + (1-p) \cdot \frac{Z(\phi_{NE}, m-k)}{Z(\phi_{NE})} \cdot \phi_{NE}^{d(\sigma,\tilde{\sigma})}}_{:=\mu}
\end{aligned}
$$

Here $Z(\phi, m-k) = \sum_{\tau:[m-k] \to [m-k]} \phi^{d(\tau,\tau^\star)}$. Therefore, $g(\sigma \mid \sigma') \geq \kappa$ for any $\sigma, \sigma'$.

We will first prove a multiplicative concentration inequality on the estimates $\widehat{g}(\cdot \mid \sigma)$ for any $\sigma$. Now fix any $\sigma$. By using lemma (1) we obtain that with probability at least $1 - \delta_1$ $\max_{\sigma'} |g(\sigma' \mid \sigma) - \widehat{g}(\sigma' \mid \sigma)| \leq \frac{\log(1/\delta_1)}{n^2}$. This implies that $\widehat{g}(\sigma' \mid \sigma) \geq g(\sigma' \mid \sigma) - \frac{\log(1/\delta_1)}{n^2}$. Since $g(\sigma' \mid \sigma) \geq \mu$, in order to have $\widehat{g}(\sigma' \mid \sigma) \geq (1-\varepsilon)g(\sigma' \mid \sigma)$ it is sufficient to have $n \geq \sqrt{\frac{\log(1/\delta_1)}{\varepsilon \cdot \mu}}$. Moreover, because of the fact that $\mu < 1$, we also have $\widehat{g}(\sigma' \mid \sigma) \leq (1+\varepsilon)g(\sigma' \mid \sigma)$. Finally, we can use union bound over all $k!$ permutations $\sigma$ and substituting $\delta_1 = \delta/(2 \cdot k!)$ we obtain that as long as the number of samples from each permutation $\sigma$ is at least $\sqrt{\frac{k \log(2k/\delta)}{\varepsilon \cdot \mu}}$, we have

$$\mathrm{Pr}\left(\forall \sigma, \sigma', \ (1-\varepsilon)g(\sigma' \mid \sigma) \leq \widehat{g}(\sigma' \mid \sigma) \leq (1+\varepsilon)g(\sigma' \mid \sigma)\right) \geq 1 - \frac{\delta}{2}$$

We now apply a similar concentration inequality for the frequency terms $f(\cdot)$. Since $\mathrm{Pr}_s(\sigma \mid \sigma^\star) \geq \mu$ for any $\sigma$ we have $f(\sigma) \geq \kappa$. By an argument very similar to the previous paragraph we have that as long as $n \geq \sqrt{\frac{\log(2/\delta)}{\varepsilon \cdot \mu}}$, we are guaranteed that $(1-\varepsilon)f(\sigma) \leq \widehat{f}(\sigma) \leq (1+\varepsilon)f(\sigma)$ with probability at least $1 - \delta/2$.

Now we provide a lower bound on the estimate $\widehat{V}(\sigma^\star)$.

$$\widehat{V}(\sigma^\star) = \widehat{f}(\sigma^\star) \sum_{\sigma'} \frac{\widehat{g}(\sigma' \mid \sigma^\star)}{\widehat{g}(\sigma^\star \mid \sigma')} \geq \frac{(1-\varepsilon)^2}{(1+\varepsilon)} f(\sigma^\star) \sum_{\sigma'} \frac{g(\sigma' \mid \sigma^\star)}{g(\sigma^\star \mid \sigma')} = \frac{(1-\varepsilon)^2}{(1+\varepsilon)} \overline{V}(\sigma^\star)$$

We now provide an upper bound on $\widehat{V}(\tau)$ for any $\tau \neq \sigma^\star$.

$$\widehat{V}(\tau) = \widehat{f}(\tau) \sum_{\sigma'} \frac{\widehat{g}(\sigma' \mid \tau)}{\widehat{g}(\tau \mid \sigma')} \leq \frac{(1+\varepsilon)^2}{(1-\varepsilon)} f(\tau) \sum_{\sigma'} \frac{g(\sigma' \mid \tau)}{g(\tau \mid \sigma')} = \frac{(1+\varepsilon)^2}{(1-\varepsilon)} \overline{V}(\tau)$$

We now use lemma ([2](#)) i.e. $\overline{V}(\sigma^\star) \geq 2\overline{V}(\tau)$.

$$\widehat{V}(\sigma^\star) \geq \frac{(1-\varepsilon)^2}{(1+\varepsilon)}\overline{V}(\sigma^\star) \geq \frac{2(1-\varepsilon)^2}{(1+\varepsilon)}\overline{V}(\tau) \geq \frac{2(1-\varepsilon)^3}{(1+\varepsilon)^3}\widehat{V}(\tau) > \widehat{V}(\tau)$$

as long as $\varepsilon \leq \frac{\sqrt[3]{2}-1}{\sqrt[3]{2}+1} \approx 0.115$. We substitute $\varepsilon = 0.1$. Finally, observe that we pick the outcome with highest empirical prediction normalized vote $\widehat{V}(\sigma)$ and with probability at least $1-\delta$, the empirical prediction normalized vote of $\sigma^\star$ will be the highest, and will be picked as the outcome. $\qquad\square$

**Lemma 1** (Theorem 9 of [7]). *Let $X_1,\ldots,X_n$ be $n$ be i.i.d. drawn from a discrete distribution $p = (p_1,\ldots,p_k)$ and let $\widehat{p}_j = \frac{1}{n}\sum_{i=1}^n \mathbf{1}\{X_i = j\}$. Then we have*

$$\Pr\left(\max_{j\in[k]}|\widehat{p}_j - p_j| \geq \frac{\log(1/\delta)}{n^2}\right) \leq \delta$$

## H.2  Separation Lemma

**Lemma 2.** *Assume uniform prior and assumption [1](#) holds. Then for any ground truth $\sigma^\star$ over subset $T$ of size $k$ and any $\tau$ with $d(\tau,\sigma^\star) \geq 1$ we have, $\overline{V}(\sigma^\star) \geq 2\overline{V}(\tau)$.*

*Proof.* Using the definition of $g(\cdot\mid\cdot)$ we can establish the following lower and upper bounds.

$$g(\sigma\mid\sigma') = \sum_{\tilde{\sigma}}\Pr_g(\tilde{\sigma}\mid\sigma')\Pr_s(\sigma\mid\tilde{\sigma}) \geq \min_{\tilde{\sigma}}\Pr_s(\sigma\mid\tilde{\sigma})\sum_{\tilde{\sigma}}\Pr_g(\tilde{\sigma}\mid\sigma') = \min_{\tilde{\sigma}}\Pr_s(\sigma\mid\tilde{\sigma})$$

$$g(\sigma\mid\sigma') = \sum_{\tilde{\sigma}}\Pr_g(\tilde{\sigma}\mid\sigma')\Pr_s(\sigma\mid\tilde{\sigma}) \leq \sum_{\tilde{\sigma}}\Pr_s(\sigma\mid\tilde{\sigma})\sum_{\tilde{\sigma}}\Pr_g(\tilde{\sigma}\mid\sigma') = \sum_{\tilde{\sigma}}\Pr_s(\sigma\mid\tilde{\sigma})$$

Now we can establish the following lower and upper bounds on the prediction-normalized vote.

$$\overline{V}(\sigma) = f(\sigma)\sum_{\sigma'}\frac{g(\sigma'\mid\sigma)}{g(\sigma\mid\sigma')} \leq \frac{f(\sigma)}{\min_{\tilde{\sigma}}\Pr_s(\sigma\mid\tilde{\sigma})}\sum_{\sigma'}g(\sigma'\mid\sigma) = \frac{f(\sigma)}{\min_{\tilde{\sigma}}\Pr_s(\sigma\mid\tilde{\sigma})}$$

$$\overline{V}(\sigma) = f(\sigma)\sum_{\sigma'}\frac{g(\sigma'\mid\sigma)}{g(\sigma\mid\sigma')} \geq \frac{f(\sigma)}{\sum_{\tilde{\sigma}}\Pr_s(\sigma\mid\tilde{\sigma})}\sum_{\sigma'}g(\sigma'\mid\sigma) = \frac{f(\sigma)}{\sum_{\tilde{\sigma}}\Pr_s(\sigma\mid\tilde{\sigma})}$$

Now consider a partial ranking $\tau$ such that $d(\tau,\sigma^\star) = 1$. Then we have,

$$\overline{V}(\sigma^\star) \geq \frac{f(\sigma^\star)}{\sum_{\tilde{\sigma}}\Pr_s(\sigma^\star\mid\tilde{\sigma})} = \frac{\Pr_s(\sigma^\star\mid\sigma^\star)}{\sum_{\tilde{\sigma}}\Pr_s(\sigma^\star\mid\tilde{\sigma})}$$

and

$$\overline{V}(\tau) \leq \frac{f(\tau)}{\min_{\tilde{\sigma}}\Pr_s(\tau\mid\tilde{\sigma})} = \frac{\Pr_s(\tau\mid\sigma^\star)}{\min_{\tilde{\sigma}}\Pr_s(\tau\mid\tilde{\sigma})}$$

Under the assumption of uniform prior we have,

$$\begin{aligned}
\Pr_s(\sigma^\star\mid\tilde{\sigma}) &= \sum_{\pi:\pi\triangleright\tilde{\sigma}}\frac{\Pr(\pi)}{\Pr(\tilde{\sigma})}\Pr_s(\sigma^\star\mid\pi)\\
&= \frac{1}{(m-k)!}\sum_{\pi:\pi\triangleright\tilde{\sigma}}\Pr_s(\sigma^\star\mid\pi)\\
&= \frac{1}{(m-k)!}\sum_{\pi:\pi\triangleright\tilde{\sigma}}\sum_{\pi':\pi'\triangleright\sigma^\star}\Pr_s(\pi'\mid\pi)\\
&= \frac{1}{(m-k)!}\sum_{\pi:\pi\triangleright\tilde{\sigma}}\sum_{\pi':\pi'\triangleright\sigma^\star}p\cdot\frac{\phi_E^{d(\pi',\pi)}}{Z(\phi_E)} + (1-p)\cdot\frac{\phi_{NE}^{d(\pi',\pi)}}{Z(\phi_{NE})}\\
&= \frac{c(T)}{(m-k)!}\left(p\cdot\frac{\phi_E^{d(\tilde{\sigma},\sigma^\star)}}{Z(\phi_E)} + (1-p)\cdot\frac{\phi_{NE}^{d(\tilde{\sigma},\sigma^\star)}}{Z(\phi_{NE})}\right)
\end{aligned}$$

Here $c(T)$ is a constant depending only on the subset $T$. Using the above identity we obtain the following lower bound on $\overline{V}(\sigma^\star)$.

$$\overline{V}(\sigma^\star) \geq \frac{p \cdot \frac{1}{Z(\phi_E)} + (1-p) \cdot \frac{1}{Z(\phi_{NE})}}{\sum_{\tilde{\sigma}} p \cdot \frac{\phi_E^{d(\tilde{\sigma},\sigma^\star)}}{Z(\phi_E)} + (1-p) \cdot \frac{\phi_{NE}^{d(\tilde{\sigma},\sigma^\star)}}{Z(\phi_{NE})}}$$

We can also obtain the following upper bound on $\overline{V}(\tau)$.

$$\overline{V}(\tau) \leq \frac{p \cdot \frac{\phi_E}{Z(\phi_E)} + (1-p) \cdot \frac{\phi_{NE}}{Z(\phi_{NE})}}{\min_{\tilde{\sigma}} p \cdot \frac{\phi_E^{d(\tilde{\sigma},\tau)}}{Z(\phi_E)} + (1-p) \cdot \frac{\phi_{NE}^{d(\tilde{\sigma},\tau)}}{Z(\phi_{NE})}}$$

We now use the relationship $p < (1-p)$ and $\phi_E < \phi_{NE}$ to improve the bounds. We will also write $Z(\phi, k) = \sum_{\tilde{\sigma}} \phi^{d(\tilde{\sigma},\tau)}$.

$$\overline{V}(\sigma^\star) \geq \frac{\frac{2p}{Z(\phi_{NE})}}{\frac{2(1-p)Z(\phi_{NE},k)}{Z(\phi_E)}} = \frac{p}{1-p} \frac{Z(\phi_E)}{Z(\phi_{NE})} \frac{1}{Z(\phi_{NE},k)}$$

$$\overline{V}(\tau) \leq \frac{2 \cdot \frac{(1-p)\phi_{NE}}{Z(\phi_E)}}{2p \cdot \frac{\phi_E^{k(k-1)/2}}{Z(\phi_{NE})}} = \frac{1-p}{p} \frac{Z(\phi_{NE})}{Z(\phi_E)} \phi_E^{k(k-1)/2}$$

Therefore, in order to have $\overline{V}(\sigma^\star) \geq 2\overline{V}(\tau)$ we need the following inequality to hold.

$$\frac{p}{1-p} \frac{Z(\phi_E)}{Z(\phi_{NE})} \frac{1}{Z(\phi_{NE},k)} \geq 2 \cdot \frac{1-p}{p} \frac{Z(\phi_{NE})}{Z(\phi_E)} \phi_E^{k(k-1)/2}$$

$\square$

# I   Screenshots from our MTurk Survey

Here, we provide screenshots of different phases of our MTurk survey.

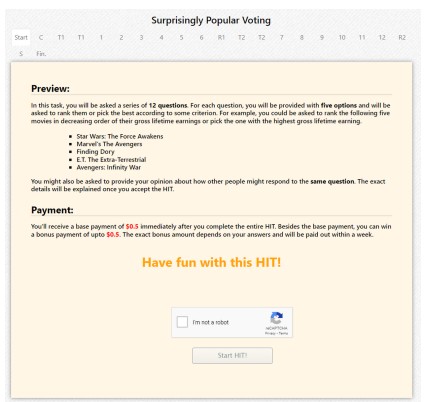 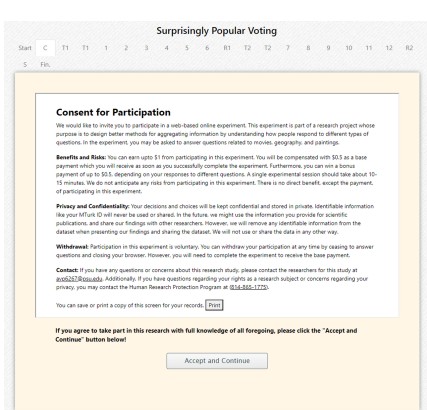

Figure 17: Preview and Consent Form

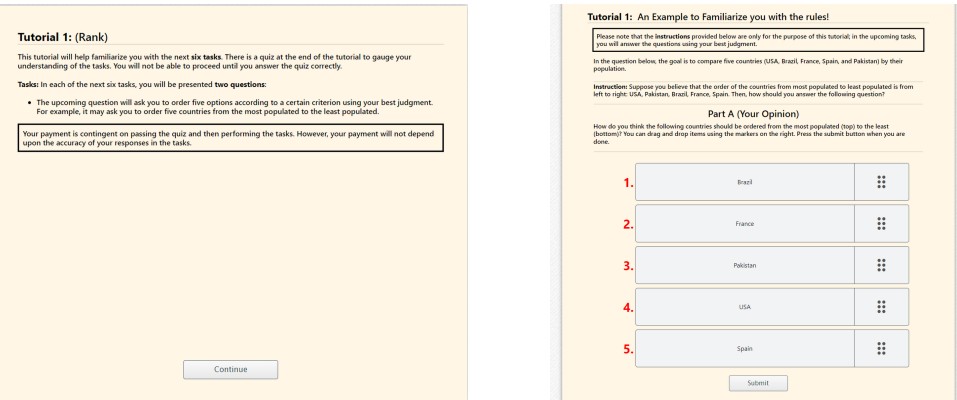

Figure 18: Tutorial for Rank-None Elicitation Format

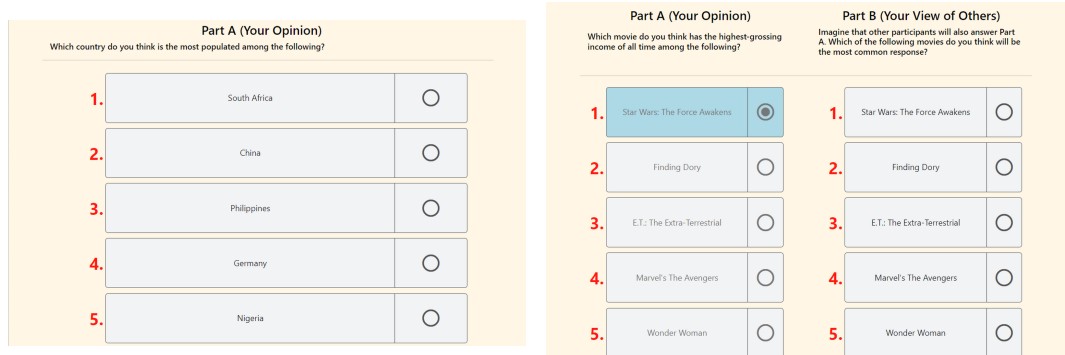

Figure 19: Questions for Top-None and Top-Top Elicitation Format

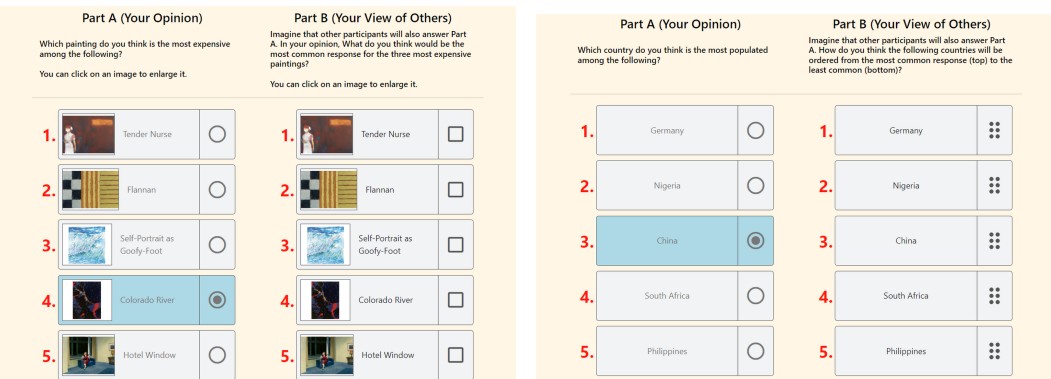

Figure 20: Questions for Top - Approval(3) and Top-Rank Elicitation Format

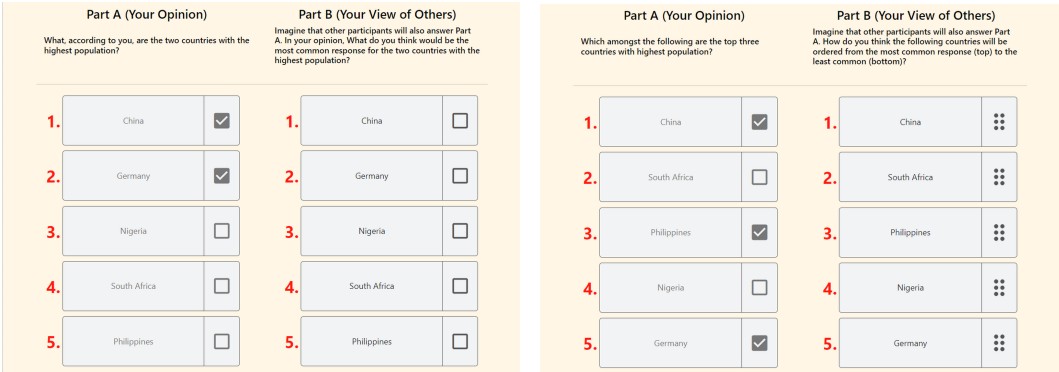

Figure 21: Questions for Approval(2) - Approval(2) and Approval(3) - Rank Elicitation Format

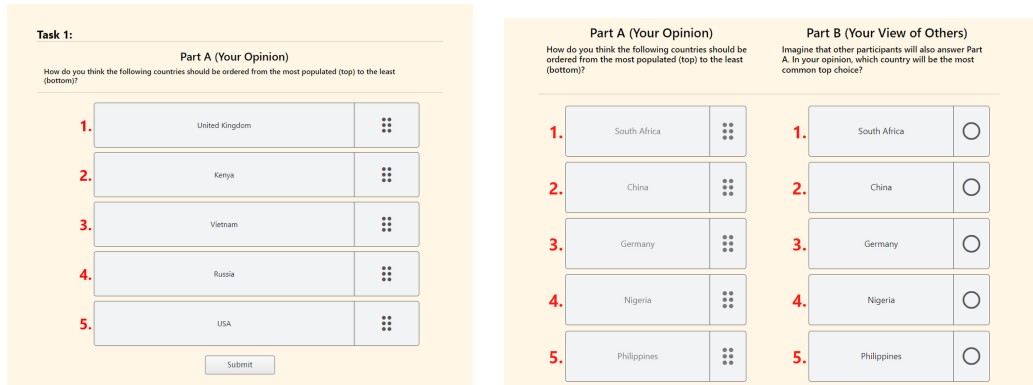

Figure 22: Questions for Rank-None and Rank-Top Elicitation Format

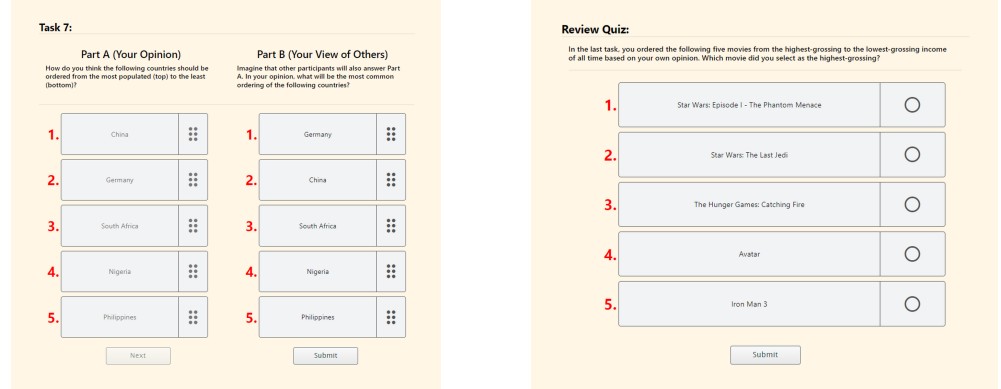

Figure 23: Question for Rank-Rank Elicitation Format and Survey Questions

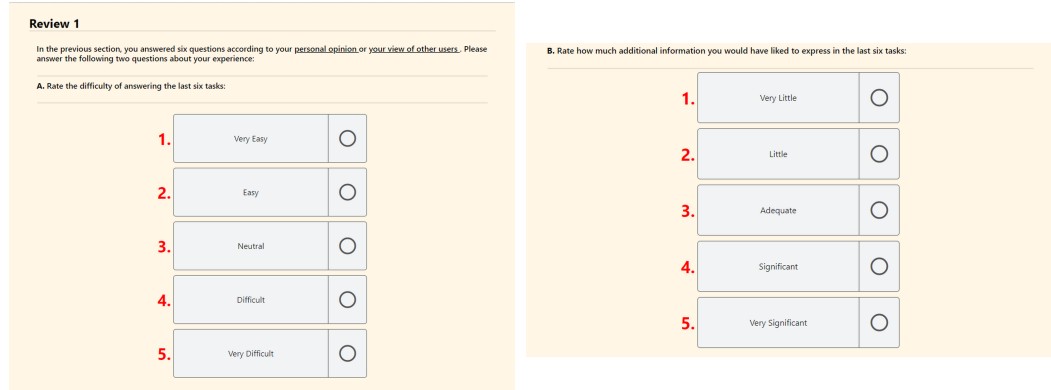

Figure 24: Difficulty and Expressiveness Questions

