# OpenReview forum: "The Surprising Effectiveness of SP Voting with Partial Preferences"
_NeurIPS.cc/2024/Conference — NeurIPS 2024 poster_

### Official Review · Reviewer_227u · 2024-06-23

**Soundness:** 2
**Presentation:** 1
**Contribution:** 3
**Rating:** 5
**Confidence:** 3

**Summary:**

Surprisingly popular voting allows to value expertise by eliciting both judgments and predictions over others' judgments. Its application to ranking is however challenged by the combinatorial size of reporting predictions over others' judgments. This paper provides practical solutions to adapt surprisingly popular voting to sparser elicitations. In particular, the authors apply this to MTurk's inputs, and show that their solutions outperform classical voting solutions. Theoretical guarantees of ground-truth recovery are also provided.

**Strengths:**

The paper provides original solutions to generalize surprisingly popular (SP) voting to sparse elicitation, which is strongly needed in many practical applications. As such, they address an important problem, with effective results.

Interestingly, the paper actually provides two solutions, and empirically show that one clearly outperforms the other (and classical voting). This is a valuable insight for this research question.

**Weaknesses:**

I found the paper quite hard to read. Key comprehension elements, such as how sparse elicitations are turned into pairwise voting inputs, or the principles of the new algorithms, are poorly explained in the main text. Even Appendix D is hard to parse (what if $a$ and $b$ are both approved? What if none is selected? Do we consider $v_i^{(a,b)}$ undefined?), with many typos (in Algorithm 2 and Algorithm 3, I don't understand what $Q$, $Q_j$, $G$ and $GT$ stand for, and how they differ).

The simulations given concentric mixtures of Mallows models also yields good insight. However, I am not sure they are convincing, because of the fact that non-expert's rankings are statistically centered around the ground truth as well. If so, this greatly undermines the story-telling of the paper, which is about valuing expertise (through SP).

The theoretical guarantees seem far from tight.

I am not quite convinced by the subset selection (line 177).

There are many imprecisions in the text:
- Line 46: "will provably recover" => it should be stressed that this depends on many assumptions. Typically, SP requires Bayesian agents, with a common prior, which update correctly based on their signals, report truthfully their vote and are numerous to avoid statistical fluctuations.
- Line 307: "using Bayesian inference" => As I understand, what was used was a MCMC sampling of the posterior, which only has statistical asymptotical properties. This is a far cry from Bayesian inference.
- Line 131: "definition 1" => Not sure what this refers to. Do you mean "equation (1)" ?
- Line 138: "definition 3" => Not sure what this refers to. Do you mean "equation (3)" ?

**Questions:**

Do I understand correctly that, in Partial-SP, we compute for each voter a set of pairwise comparisons $a \succ b$, each of which unbiases non-expertise with SP? And that the resulting voter-wise sets of unbiased comparisons are aggregated through a voting rule?

Corollary 1 says $n \geq \Omega(k!)$ inputs are needed *for all subsets of $S$*. Does this imply that the total number of inputs must be exponential in $S$?

More generally, I am surprised that, in the proposed Mallows models, $k!$ inputs are needed. I would have guessed that $k^2$ are sufficient, as it suffices to be correct on all comparisons of pairs of alternatives. I even feel that this insight should hold for SP with a non-expert ranking distribution that is not centered on $\pi^\star$. Would the authors agree?

First, what value $s$ is chosen? More importantly, it feels that two human subjects will either likely have zero or many (around k/2) alternatives to compare in common. Why would this be preferable to random subsets? What about minimizing the number of pairs of subjects that have $< 2$ alternatives in common?

I am quite open to increasing my rating, depending on the authors' response.

**Limitations:**

One limitation that failed to be highlighted is the dependency of the new algorithms on hyperparameters $\alpha$ and $\beta$. Not mentioning this is misleading as it suggests that the new algorithms are less arbitrary than they actually are.

I am disturbed by the conclusion on "political polling or collective moderation of online content". While expertise is clearly lacking and not uniformly distributed, it is questionable the extent to which SP (and variants) should be applied to such issues, where there is arguably no ground truth (formally, there is no "common prior" of the SP model). I would have appreciated adding caveats, and perhaps a mention of the (exciting) social science / philosophy questions that generalizing SP to such topics would raise.

---

> ### Author Rebuttal · Authors · 2024-08-06
>
> We thank the reviewer for the valuable feedback. They are very helpful and we will incorporate them in the next version of the paper.
>
> > “Do I understand correctly that, in Partial-SP, we compute for each voter a set of pairwise comparisons $a\succ b$ , each of which unbiases non-expertise with SP? ...”
>
> The right way to think about Partial-SP is the following: for each subset (not voter), we compute a set of pairwise comparisons a>b after collecting votes from the voters. Then SP unbiases non-expertise to give us unbiased ranking over the subsets (each of size $k$). Then the resulting partial rankings are aggregated through a voting rule to generate a ranking over the $m$ alternatives.
>
> > “Corollary 1 says $n\ge \Omega(k!)$ inputs are needed for all subsets of S. Does this imply that the total number of inputs must be exponential in S?”
>
> Corollary 1 uses theorem 1 for the sample complexity results which derives bounds for a naive version of the SP algorithm. Theorem 1 treats each of the k! permutations as possible ground truths and picks the ranking with largest prediction-normalized score. The sample complexity is approximately $O(G \sqrt{\log G})$ where G is the number of possible ground truths. For $G=k!$ we get the sample complexity bound of $\tilde{O}(k! \sqrt{k})$. It is straightforward to consider a version that applies SP-voting to each pairwise preference and then aggregates them. For a pairwise preference, $G = 2$ and as long as $n \ge O(\sqrt{\log(k/\delta)}$ SP-voting can recover the pairwise preference with prob at least $1-\delta/k^2$. Therefore, by a union bound, the total number of samples required to recover the ground truth over a subset of size k is $\tilde{O}(k^2)$.
>
> > “More generally, I am surprised that, in the proposed Mallows models, k! inputs are needed...”
>
> As mentioned above, the bound of $\tilde{O}(k!)$ is needed only when one applies a naive version of the SP algorithm. The reviewer is right that the total sample complexity can be reduced to $O(|S| \cdot k^2 \log (|S| k / \delta) )$ when one applies SP-voting to each pair within a subset. However, even with the naive version of the SP algorithm the total sample complexity $\tilde{O}(|S| \cdot k!)$ can be better than $m^2$ when $|S| = O(m)$ and $k = o(\log m)$. This is precisely our setting since the subset size $k$ is $5$ or $6$ and the total number of alternatives in the ground truth is $m = 35$, and hence $k! < m^2$.
>
> > “First, what value s is chosen? … Why would this be preferable to random subsets? ...”
>
> We chose stepsize $s$ to be $6$. Through simulation, we observed that increasing subset size ($k$) and step size ($s$) results get better but step size $6$ was a fair number to balance cognitive load and also create overlaps between subsets.
> The main idea is to choose subsets that have alternatives in common. Then we can estimate the partial rankings of each subsets and use transitivity of rankings to infer a global ranking over the $m$ alternatives. If we choose subsets uniformly at random then we will need a lot of subsets (and hence a lot of voters) before we have high overlap among different subsets. This is the reason we fixed the choice of the subsets and then determined assignments to voters through an integer programming which guarantees that all the voters report approximately an equal number of comparisons.
>
> > “One limitation that failed to be highlighted is the dependency of the new algorithms on hyperparameters $\alpha$ and $\beta$ ...”
>
> We have highlighted the dependency of the new algorithms on hyperparameters $\alpha$ and $\beta$ in the appendix (line 662). Using grid search, we observed that as long as $\alpha > 0.5$ and $\beta < 0.5$, the new algorithms recover similar results. This means that the algorithms are robust to the choices of the hyperparameters $\alpha$ and $\beta$.
>
> > “I am disturbed by the conclusion on "political polling or collective moderation of online content"...”
>
> Thank you for raising the issue of extending SP to social science/philosophy questions! We would like to point out that SP is already being used in a couple of applications.
>
> - Polling opinions in political elections e.g. by CBC in Canadian elections.
> - Some conferences have been experimenting with SP type methods for ranking papers by asking reviewers to score a paper and also provide a prediction of what they think scores of others will be (EC and ICML this year).
>
> The main difficulty is that most of the questions in these domains are inherently subjective with no ground truth. Therefore, it is difficult to determine the expertise of a voter and more research is needed to determine what should be the right framework for differentiating voters.
>
>
> > “The simulations given concentric mixtures of Mallows models also yields good insight. However, I am not sure they are convincing…”
>
> - Yes, the non-expert’s rankings are statistically centered around the ground truth, however, note that their distance from the ground truth ranking is around 0.7 in terms of average Kendall-Tau distance. There was a minor bug in our code and the new plots are attached with the rebuttal, see figures 1 and 2. An alternative would be a mixture model with two centers where experts and non-experts report close to their own centers (respectively $\pi^\star_E$, and $\pi^\star_{\textrm{NE}}$). However, we believe this is not the right model for our setting as there is already an objectively correct ranking over all the alternatives.
>
> - Furthermore, to demonstrate that the second type of mixture model is unsuitable for our setting, we conducted a comparative analysis. We selected ground-truth as centers for experts, while for non-experts, we chose random centers within a [0,1] range of Kendall's tau distance from the ground truth. Figure 3 in our NeurIPS Rebuttal plot illustrates the resulting poor fit, highlighting a significant lack of overlap between the non-expert distributions in synthetic data compared to real data.

---

> > ### Comment · Reviewer_227u · 2024-08-09
> >
> > I thank the authors for their helpful rebuttal. While I believe that the paper has still a lot of room for improvements in terms of writing, proofs (by replacing $k!$ by $k^2$) and synthetic experiments (by a more careful analysis of non-centered non-experts), I have upgraded my rating to 5.

---

> > > ### Comment · Reviewer_227u · 2024-08-12
> > >
> > > Out of personal curiosity, the authors wrote that CBC in Canadian elections used the surprisingly popular vote. I would be very interested to have the source of the information (or of similar political uses of the vote).

---

> > > > ### Author Response · Authors · 2024-08-13
> > > >
> > > > We could not find the source for the CBC's article but a similar and relevant article titled - "Election Polls Are More Accurate if They Ask Participants How Others Will Vote", can be found online where they talk about using the surprisingly popular method in Polling. Due to guidelines, we cannot directly share the link here but we'd be happy to furnish more details about this source if needed.

---

> > > > > ### Comment · Reviewer_227u · 2024-08-13
> > > > >
> > > > > Thank you very much. This is very interesting.

---

### Official Review · Reviewer_GckY · 2024-07-13

**Soundness:** 4
**Presentation:** 3
**Contribution:** 2
**Rating:** 7
**Confidence:** 3

**Summary:**

The paper generalizes the surprisingly popular algorithm to the partial ranking setting. The prior method that considers this generalization only works when every agent provides her signal over the full ranking. The current paper considers how to elicit only partial information from agents and aggregate them using SP. The proposed method can scale with the number of items. Furthermore, the paper proposes a model to capture the responses from human agents, which can be used for theoretical analysis.

**Strengths:**

* Generalizing existing methods for broader real-world applications is very necessary. The paper, overall, is making non-trivial progress in this direction.
* The paper proposes a theoretically trackable model to capture agents’ behaviors while reporting ranking information. If calibrated (I think we should be careful to claim calibration here, see weakness), the model can be useful for future research on ananlyzing crowdsourcing of ranking data.
* The experiments are robust: tested on three datasets, measured using various methods, and compared with multiple baselines.

**Weaknesses:**

* First of all, the prior paper [25] seems to greatly discount the contribution (and effort) of this work. In particular, they use the same datasets, very similar experimental designs, and the same model. However, the current paper only speaks about the high-level differences between these two papers. In particular, as stated in line 98, [25] seems to elicit pairwise comparison from agents and then aggregate them, which is pretty much the proposed method Partial-SP. The only difference I can see is that Partial-SP allows k partial rankings while [25] only uses 2. I could be wrong about [25], but the authors should have a more detailed comparison. Also, it doesn’t seem the method in [25] is considered as a baseline. Why not?
* In line 310, “As seen 310 from the posterior distributions for the dispersion parameters in Figure 5 synthetic data generation 311 process accurately replicates real data characteristics”. I’m not sure how “accurate this is as the distributions for non-expert seem to be way off. How confident can we trust the considered model for downstream theoretical analysis?
* In line with the previous comment, is the same dataset used to fit the model, then compared with the simulated data as shown in Figure 5? Or was the dataset separated into a training set and a test set? If it’s the former, how do we know if Figure 5 is not overfitting the data? A lot of the content is left in the Appendix, but some of it is necessary to understand the results, so it might be better to have it back.

**Questions:**

See weakness

**Limitations:**

Limitations are fairly discussed

---

> ### Author Rebuttal · Authors · 2024-08-06
>
> We thank the reviewer for insightful feedback. Below we provide detailed responses to the questions.
>
> > “First of all, the prior paper [25] seems to greatly discount the contribution (and effort) of this work. In particular, they use the same datasets, very similar experimental designs, and the same model.”
>
> - We would like to highlight that our work is significantly different than [25] and not a natural extension of prior work. [25] is interested in recovering a ground truth over (m=4) alternatives by presenting these same alternatives to all voters and asking for various reports. We, on the other hand, are interested in recovering ground truth over a large number of alternatives ($m\sim 30$) by asking reports on smaller subsets ($k=5,6$). This introduces two new challenges – design of new elicitation formats for partial preferences (e.g. Approval(t)-Rank as mentioned in line 151), and figuring out how to extend SP-voting for handling partial preferences.
>
> - Although we adopt a randomized experimentation framework similar to [25], there are two differences. First, we need to determine which subsets (of size $k\ll m$) we should use to elicit partial information. Having too many subsets will blow up the sample complexity, and having too few (or non-overlapping) subsets will render the problem impossible. Besides, as mentioned above, we also consider the design of new elicitation formats suitable for partial reports.
>
> - We don’t use the dataset used by [25], and in fact, the dataset is not even applicable for our setting. This is because, in their experiment, the voters provide reports (vote or prediction) under the instruction that the considered set of 4 alternatives are the only alternatives in the ground truth. Moreover, the subsets in the datasets are non-overlapping and cannot help us recover a ground truth over the $m$ alternatives.
>
> > “In particular, as stated in line 98, [25] seems to elicit pairwise comparison from agents and then aggregate them, which is pretty much the proposed method Partial-SP. The only difference I can see is that Partial-SP allows k partial rankings while [25] only uses 2.”
>
> For handling partial preferences, we propose two methods – Partial-SP and Aggregated-SP. The first method, Partial-SP goes beyond just eliciting pairwise comparisons and aggregating them. The main idea is to first determine the correct partial rank for each subset (by eliciting any type of information be it approval, rank, etc.) and then aggregating them through a voting rule. As discussed in the introduction, applying the basic version of SP-voting requires eliciting $O(m^2)$ pairwise information which can be large. Hence the generalization of $k > 2$ is important and non-trivial. Besides, partial-SP applies SP-voting at each subset level and then aggregates them through a voting rule, and in that sense, it is significantly different than the prior work [25]. Furthermore, the second algorithm, Aggregated-SP, offers a different perspective on handling partial preferences in the SP voting framework. In lines 161-176 we only provide an outline of the two algorithms because of limited space, but we can definitely move some details from the appendix.
>
> > “Also, it doesn’t seem the method in [25] is considered as a baseline. Why not?”
>
> It's important to note that SP-voting [25] requires pairwise preference data for all $O(m^2)$ pairs as it builds a tournament over the m alternatives by applying SP-voting on each pair. When m is large (e.g. $m\sim 30$ for our setting) and it is impossible to collect human preferences (vote and report) for all pairs. Therefore, because of the missing data, SP-voting algorithm cannot be evaluated. Our novelty lies in generalizing SP-voting with partial preferences, eliciting O(mk) amounts of information and yet recovering the ground truth over m alternatives.
>
>
> > “ I’m not sure how “accurate" this is as the distributions for non-expert seem to be way off. How confident can we trust the considered model for downstream theoretical analysis?”
>
> - Thank you for your observation. Upon re-evaluating our parameter inference approach, we identified that the scipy.stats.kendalltau function we used computes Kendall's tau correlation, and not Kendall's tau distance. This distinction led to discrepancies in the non-expert distributions between real and synthetic data. We have addressed this by implementing the correct Kendall's tau distance metric. Figures 1 and 2 in our NeurIPS Rebuttal plots file show the updated plots with a perfect fit, particularly for the **Movie** domain (figure 1).
>
> - When combining all datasets, we observe significant overlap in individual expert and non-expert distributions, as well as a prominent distinction between expert and non-expert groups in terms of Kendall's tau distance. Although there is some mismatch between the original data and synthetic data for the non-experts, we believe this is because of a limited number of samples, and heterogeneity of different domains.
>
>
> > “In line with the previous comment, is the same dataset used to fit the model, then compared with the simulated data as shown in Figure 5? Or was the dataset separated into a training set and a test set? If it’s the former, how do we know if Figure 5 is not overfitting the data? “
>
> Yes, we used the whole dataset to fit the model, sampled from the posterior of the fitted model, and then checked the overlap. The practice of splitting the dataset into test and training sets is common in supervised learning setting. We work in a Bayesian setting where different types of checks (e.g. posterior predictive checks) are used to determine the fitness of a model. Additionally, we have a limited number of samples, and that's another reason we used the whole dataset to fit the model.

---

> > ### Comment · Reviewer_GckY · 2024-08-11
> > **Post-rebuttal**
> >
> > I appreciate the author's rebuttal. I have raised my score.

---

### Official Review · Reviewer_TQiC · 2024-07-13

**Soundness:** 3
**Presentation:** 3
**Contribution:** 2
**Rating:** 6
**Confidence:** 2

**Summary:**

This paper studies the problem to recover the ground truth ordering over a large number of alternatives. The assumption is that the ground truth ranking is drawn from a prior, and each voter observes a noisy version of the ground truth. It was previously shown that the surprisingly popular (SP) algorithm could recover the ground truth even when experts are in minority. In this paper, the authors propose Aggregated-SP and Partial-SP, that ask voters to report vote and predictions on a subset of alternatives in terms of top alternative, partial rank, or an approval set. Experimental results show that the proposed algorithms outperform conventional preference aggregation algorithms for the recovery of ground truth rankings. The authors also provide theoretical bounds on the sample complexity of SP algorithms with partial rankings.

**Strengths:**

This paper extends previous work on the SP algorithms to only require the voters to report predictions on a subset of alternatives instead of for all the alternatives. Human generated datasets are used to evaluate the proposed algorithms. The experiments are well designed and clearly described. The results are sufficiently analyzed and discussed. Finally, sample complexity is analyzed under a Mallows model distribution assumption.

**Weaknesses:**

The theoretical assumptions in Section 6 are somehow constrained, especially Assumption 1.

**Questions:**

* In the introduction, the authors mention that the Surprisingly Popular Voting algorithm can recover the ground-truth ranking. I wonder if this is a theoretical guarantee or empirical / experimental.
* Have you compared different aggregation algorithms for other elicitation formats besides Rank-Rank as shown in Figure 4?
* Is it possible to show that recovering the true partial ranking is impossible if Assumption 1 is not true?

**Limitations:**

The authors discuss limitations and future directions in the paper, including exploring the setting of SP beyond the majority-minority dichotomy, or with malicious voters.

---

> ### Author Rebuttal · Authors · 2024-08-06
>
> We thank the reviewer for the feedback and insightful questions. Below we provide answers to the questions.
>
>
> > “In the introduction, the authors mention that the Surprisingly Popular Voting algorithm can recover the ground-truth ranking. I wonder if this is a theoretical guarantee or empirical / experimental.”
>
> Theorem 3 of Prelec et al. [36] guarantees that the Surprisingly Popular Voting (SP-Voting) can recover the ground-truth ranking with a probability of 1 as the number of voters approaches infinity (As written in line 46 in our paper). Note that this guarantee holds only with infinite data, and to the best of our knowledge, theorem 1 in our paper is one of the first results providing finite-sample guarantees for SP-type algorithms.
>
> > “Have you compared different aggregation algorithms for other elicitation formats besides Rank-Rank as shown in Figure 4?”
>
> Yes, we have compared different aggregation algorithms for all the elicitation formats used in our study and have shown the results in Figure 9 and Figure 10 of Appendix G.3.
>
>
> > “Is it possible to show that recovering the true partial ranking is impossible if Assumption 1 is not true?”
>
>
> - If Assumption 1 is not true, then either $p$ (fraction of experts) is very small or the dispersion parameter of the non-experts ($\phi_\textrm{NE}$) is very large. First, suppose $\phi_{\textrm{NE}}$ is large (e.g. arbitrarily close to $1$) then the distribution $Pr_s(\pi | \pi^\star, \phi_{\textrm{NE} })$ is almost a uniform distribution and has no information about the true rank $\pi^\star$. In such a scenario if $p$ is also very small ($\approx 0$), then the signal distribution $Pr_s(\pi_i | \pi^\star)$ is almost a uniform distribution, and, the prediction distribution of both the experts and the non-experts are similar. Both will predict approximate uniform distribution, and it is impossible to recover the true ranking. Once either $p$ increases or $\phi_{\textrm{NE}}$ decreases, then either the prediction distribution of the experts will change ($p$ bounded away from $0$) or the signal distribution will be concentrated around $\pi^\star$ (when $ \phi_{\textrm{NE}} \ll 1$), making identification possible.
>
> - We would like to point out that for applying the SP algorithm to more than two alternatives, Prelec et. al. [36] also required a condition on the underlying data generating process ($P(v_i|s_i) > P(v_i|s_j)$). Our condition is adapted for the specific case of concentric Mallows model, and implies their condition. We believe it might be possible to obtain a better trade-off between the parameters $p$ and $\phi_{\textrm{NE}}$ through a more sophisticated analysis, but a condition specifying the trade-off between the two parameters is necessary for identifying the true rank.

---

> > ### Comment · Reviewer_TQiC · 2024-08-12
> >
> > I thank the authors for their response. I have raised my score.

---

### Official Review · Reviewer_1pvg · 2024-07-19

**Soundness:** 4
**Presentation:** 4
**Contribution:** 4
**Rating:** 7
**Confidence:** 4

**Summary:**

The paper extends the previous work of (Hosseini et al. 25) by overcoming a major weakness: eliciting a full ranking and a prediction about the ranking is too costly. The paper designed a method that elicit partial preferences and recover the full ranking by aggregating partial rankings. They empirically test the method through a large-scale crowdsourcing experiment on MTurk and show their approaches outperform conventional preference aggregation algorithms for the recovery of ground truth rankings. They analyze the collected data and demonstrate that voters' behavior in the experiment, including the minority of the experts, and the SP phenomenon, can be correctly simulated by a concentric mixtures of Mallows model. Finally, they provide theoretical bounds on the sample complexity of the proposed methods.

**Strengths:**

The paper tackles an intriguing problem and addresses a significant flaw in previous approaches: the impracticality of eliciting full rankings and making predictions about those rankings in real-world scenarios. The authors develop simple methods to mitigate this issue by eliciting and aggregating partial rankings. They test the effectiveness of their approach through real-world experiments and introduce a novel explanation of voters' behavior based on the data. I see the experimental results as the primary contribution of the work, given that previous literature on the topic has been largely theoretical. It is particularly nice that the authors apply their method in practical settings and prove its effectiveness. Additionally, their analysis of sample complexity offers a valuable theoretical contribution.

**Weaknesses:**

The primary technique follows the methodology of Hosseini et al. (25), with a relatively simple extension. Nonetheless, I do not see the technical contribution as the most important contribution of the work.

**Questions:**

None.

**Limitations:**

Yes.

---

> ### Author Rebuttal · Authors · 2024-08-06
>
> Dear reviewer, thank you for your feedback and insightful comments.  If you have additional questions, do let us know and we will be happy to answer them.

---

### Author Rebuttal · Authors · 2024-08-06

Dear reviewers,

Many thanks for your feedback and insightful comments. Upon re-evaluating our parameter inference approach, we identified that the scipy.stats.kendalltau function we used computes Kendall's tau correlation, and not Kendall's tau distance. This distinction led to discrepancies in the non-expert distributions between real and synthetic data. We have addressed this by implementing the correct Kendall's tau distance metric.

- Figures 1 and 2 in our NeurIPS Rebuttal plots file show the updated plots with a perfect fit, particularly for the Movie domain. When combining all datasets, we observe significant overlap in individual expert and non-expert distributions, as well as a more prominent distinction between expert and non-expert groups in terms of Kendall's tau distance. In fact, the votes of non-experts are now centered around 0.7 which is further away from the experts' votes.

- Figure 3 compares original data and synthetic data when experts and non-experts have different centers. However, we observed that model fit is poor for non-expert groups.

---

### Decision · Program_Chairs · 2024-09-25

**Decision:**

Accept (poster)

**Comment:**

The reviewers were initially mixed on this paper, but throughout the rebuttal process they were raised significantly. The paper is overall well-motivated and has compelling empirical results. I had some concerns about writing quality, but the authors were responsive throughout the review process and the typos identified by reviewers are all easily fixable. The authors also concerningly had an error in one of their evaluation methods, but they claim to have fixed it. Lastly, they responded to (extremely minor) ethical considerations raised by ethics review for their use of MTurk.

Personally, I found the paper compelling. The authors' main theoretical result for sample complexity isn't the most technically complex, but the paper's main contribution is on the empirical side. The main weakness identified by reviewers was a very related paper ([25]) that only differed in the size of the partial rankings elicited (pairwise vs. k). However, the authors did a good job explaining why this paper made a meaningful contribution past [25], and they will add it to the final draft.